# EGFR-targeted fluorescence molecular imaging for intraoperative margin assessment in oral cancer patients: a phase II trial

Jaron G. de Wit[1,11], Jasper Vonk[1,11], Floris J. Voskuil [1,2],
Sebastiaan A. H. J. de Visscher [1], Kees-Pieter Schepman[1],
Wouter T. R. Hooghiemstra[3,4], Matthijs D. Linssen[3,4], Sjoerd G. Elias[5],
Gyorgy B. Halmos[6], Boudewijn E. C. Plaat [6], Jan J. Doff [2], Eben L. Rosenthal[7],
Dominic Robinson[8], Bert van der Vegt [2], Wouter B. Nagengast[4],
Gooitzen M. van Dam[9,10] & Max J. H. Witjes [1] ✉

Inadequate surgical margins occur frequently in oral squamous cell carcinoma surgery. Fluorescence molecular imaging (FMI) has been explored for intraoperative margin assessment, but data are limited to phase-I studies. In this single-arm phase-II study (NCT03134846), our primary endpoints were to determine the sensitivity, specificity and positive predictive value of cetuximab-800CW for tumor-positive margins detection. Secondary endpoints were safety, close margin detection rate and intrinsic cetuximab-800CW fluorescence. In 65 patients with 66 tumors, cetuximab-800CW was well-tolerated. Fluorescent spots identified in the surgical margin with signal-to-background ratios (SBR) of ≥2 identify tumor-positive margins with 100% sensitivity, 85.9% specificity, 58.3% positive predictive value, and 100% negative predictive value. An SBR of ≥1.5 identifies close margins with 70.3% sensitivity, 76.1% specificity, 60.5% positive predictive value, and 83.1% negative predictive value. Performing frozen section analysis aimed at the fluorescent spots with an SBR of ≥1.5 enables safe, intraoperative adjustment of surgical margins.

Oral squamous cell carcinoma (OSCC) accounts for over 375,000 new cases and 175,000 deaths year[1]. Surgical resection is often the primary treatment, with the aim of complete tumor removal with a sufficient margin of healthy tissue. In OSCC surgery, a complete resection (or R0 resection) is defined as a histological margin of ≥5 mm. Yet, inadequate

margins (<5 mm) occur in OSCC surgery at one of the highest rates in surgical oncology[2], with tumor-positive (0–1 mm) margins occurring in up to 40% of cases and close margins (1–5 mm) in up to 45%[3–6], primarily located in the deep margin[3]. An inadequate margin is a significant predictor of local recurrence and is strongly associated with

[1]Department of Oral & Maxillofacial Surgery, University of Groningen, University Medical Centre Groningen, Groningen, the Netherlands. [2]Department of Pathology & Medical Biology, University of Groningen, University Medical Centre Groningen, Groningen, the Netherlands. [3]Department of Clinical Pharmacy and Pharmacology, University of Groningen, University Medical Centre Groningen, Groningen, the Netherlands. [4]Department of Gastroenterology and Hepatology, University of Groningen, University Medical Centre Groningen, Groningen, the Netherlands. [5]Department of Epidemiology, Julius Centre for Health Sciences and Primary Care, University Medical Centre Utrecht, Utrecht University, Utrecht, the Netherlands. [6]Department of Otorhinolaryngology, Head and Neck Surgery, University of Groningen, University Medical Centre Groningen, Groningen, the Netherlands. [7]Department of Otolaryngology, Vanderbilt University Medical Centre, Nashville, Tennessee, United States of America. [8]Center for Optical Diagnostics and Therapy, Department of Otorhinolaryngology and Head and Neck Surgery, Erasmus MC Cancer Institute, Rotterdam, the Netherlands. [9]Department of Nuclear Medicine and Molecular Imaging, University of Groningen, University Medical Centre Groningen, Groningen, the Netherlands. [10]TRACER Europe B.V. / AxelaRx, Groningen, the Netherlands. [11]These authors contributed equally: Jaron G. de Wit, Jasper Vonk. ✉e-mail: m.j.h.witjes@umcg.nl

disease-specific mortality[4]. By avoiding inadequate margins during resection, surgeons can positively influence prognosis at the most crucial time[4,7,8]. In the current standard of care, the final histopathological margin status is available only several days post-surgery, so surgical correction is possible only after the initial resection. Secondary salvage surgery after previous irradical resections is associated with increased morbidity[9,10], and oncological outcomes are worse compared to initial tumor free resections[11]. In general, patients with irradical resections require adjuvant radiotherapy/chemoradiotherapy, which are associated with severe side effects[12–14].

There is a clinical need for a diagnostic tool that provides a swift intraoperative assessment of the complete resection margin, enabling surgeons to correct the margins immediately. The only widely used intraoperative technique in OSCC surgery is fresh frozen sectioning, during which only a fraction of the resection margin is analyzed due to the small number of sections that can be obtained. Fresh frozen sectioning is also affected by surgical sampling error[15–17], leading to a discordance with final histopathology of 4.3% of tumor-positive margins and 20% of close margins[16–19].

Fluorescence molecular imaging (FMI) is a promising approach to avoid this problem. FMI is a wide-field imaging technique that uses tumor-specific fluorescent tracers to enhance the visualization of tumor tissue[20,21]. In OSCC, the epidermal growth factor receptor (EGFR) has been frequently used as a target in FMI; this receptor is overexpressed in over 90% of OSCCs[22,23]. Multiple phase I feasibility studies have demonstrated the safety of EGFR-targeting tracers such as cetuximab-800CW and panitumumab-800CW, and their potential for real-time intraoperative margin assessment by showing fluorescent spots (possible lesions) in the deep margin[24–26]. In these studies, predosing with unlabeled antibody prior to tracer administration showed improved contrast, most likely by preventing rapid plasma clearance of the tracer and occupying off-target receptors in normal tissue[27,28]. To date, no well-powered phase II studies to evaluate the diagnostic accuracy of EGFR-targeted FMI for intraoperative margin assessment have been published so far.

In this phase II clinical study, we demonstrate that intraoperative specimen-driven margin assessment with FMI using cetuximab-800CW detects tumor-positive margins and close margins with high sensitivity. Therefore, this imaging approach could lead to a decreased necessity of adjuvant chemoradiotherapy when intraoperatively detected positive margins are surgically adjusted.

## Table 1 | Patient and specimen characteristics

| Patient characteristics | | |
|---|---|---|
| Age | | 68 (29–90) |
| Female | | 33 (51) |
| Radiotherapy | | 20 (30) |
| Previous surgery | | 19 (29) |
| **Tumor characteristics** | | |
| Tumor location | Tongue | 30 (45) |
| | Mandibular gingiva | 20 (30) |
| | Maxillar gingiva | 6 (9) |
| | Floor of mouth | 5 (7) |
| | Cheek | 4 (6) |
| | Buccal fold | 1 (1) |
| | Glossotonsillar sulcus | 1 (1) |
| T-stage | T1 | 24 (36) |
| | T2 | 22 (33) |
| | T3 | 4 (6) |
| | T4 | 17 (25) |
| Maximum diameter (mm) | | 22.8 (4.0–60.0) |
| Depth of invasion (mm) | | 7.4 (0.1–31.0) |
| Tumor thickness (mm) | | 7.9 (0.9–27.0) |

Data are presented as median (range) or n (%). Source data are provided as a Source Data file.

# Results

## Patient characteristics
Between January 17, 2019, and November 29, 2021, 74 patients were enrolled in this study. In 4 patients (5%), an infusion-related adverse event led to the termination of intravenous administration. All these adverse events occurred during the administration of unlabeled cetuximab (see 'Safety data'). Four other patients could not be included in the final analyses since all tumor tissue was removed during diagnostic biopsy without this being clinically evident. In one patient, all images had to be excluded (see "Methods"). In the remaining study population of 65 patients, we analyzed 66 tumor specimens since one patient presented with two separate primary tumors on both sides of the tongue. The median age of patients was 68 years (range 29–90) and 33 were female (51%). Primary tumors were located in the tongue ($n = 30$), mandibular gingiva ($n = 19$), maxillary gingiva ($n = 6$), floor of mouth ($n = 5$), cheek ($n = 4$), buccal fold ($n = 1$), and glossotonsillar sulcus ($n = 1$). Final histopathology showed 14 tumor-positive margins, 37 close margins, and 71 tumor-negative margins. In the final population, 19 patients (29%) had previously undergone surgery for other primary oral cancers, and 20 (30%) patients had previously received radiotherapy in the head and neck region. In 64/65 patients, the tracer administration was performed 2 days before surgery, and one patient received the study drugs 3 days before surgery due to logistical reasons. Patient demographics, clinical and pathological data are summarized in Table 1. The study workflow of imaging procedures, pathology processing and final histopathology is depicted in Fig. 1. Survival statistics are provided in the Supplementary Information.

## In vivo fluorescence contrast
Based on in vivo visual inspection, all tumors showed increased fluorescence signal compared to adjacent normal tissue. Median intrinsic fluorescence in tumors was significantly higher in tumor tissue ($3.3$ ($2.7$–$6.1$) $\times 10^{-2}$ mm$^{-1}$) compared to adjacent tissue ($1.0$ ($0.9$–$1.5$) $\times 10^{-2}$ mm$^{-1}$, $p = 0.0001$), and the median TBR$_{spectroscopy}$ was $3.1$ ($2.0$–$5.4$) (Fig. 2). Intraoperative in vivo fluorescence imaging of the oral cavity and the wound bed revealed three satellite lesions located peripheral from the tumor and one lesion in the intraoral wound bed after excision, all of which were clinically not suspicious for tumor tissue. Diagnostic biopsy of the satellite lesions revealed low-grade dysplasia in one and invasive tumor in the other two, leading to secondary surgery (Supplementary Fig. 1a). The lesion in the wound bed turned out to be an artery (Supplementary Fig. 1b).

## Ex vivo margin assessment
In the 66 included surgical specimens, 113 surgical planes were assessed within minutes after excision (Fig. 3). In 64 (57%) surgical planes, no fluorescent spots were identified; consequently, an SBR of 1 was reported. In the remaining 49 (43%) surgical planes, we observed 58 fluorescent spots. SBR values of all spots were compared with the histopathological margin in millimeters. The ROC curve depicted in Fig. 4 shows an area under the curve (AUC) of 0.95 (95% CI 0.91 to 1.0) for the detection of tumor-positive margins. The optimal cut-off for tumor-positive margin detection was an SBR ≥2, based on Youden's index. This provides a sensitivity of 100% (95% CI 78.5–100%), a specificity of 85.9% (95% CI 71.1–93.8%), 58.3% (95% CI 38.1–76.1%) PPV, 100% (95% CI 94.1–100%) NPV. Close margins were detected with a sensitivity of 43.2% (95% CI 28.0–59.7%), a specificity of 85.9% (95% CI 71.1–93.8%), 61.5% (95% CI 43.4–77.0%) PPV, 74.4% (95% CI 59.8–85.0%) NPV. In total, 14/14 tumor-positive margins, 16/37 close margins, and 10/73 tumor-negative margins showed an SBR ≥2.

An SBR cut-off of ≥1.5 was optimal for the detection of close margins (1–5 mm) based on Youden's index. We detected close margins with a sensitivity of 70.3% (95% CI 53.3–83.0), a specificity of 76.1% (95% CI 58.6–87.7%), 60.5% (95% CI 44.7–74.3%) PPV, 83.1% (95% CI 65.8–92.6%) NPV, and an AUC of 0.72 (95% CI 0.62–0.82). To determine

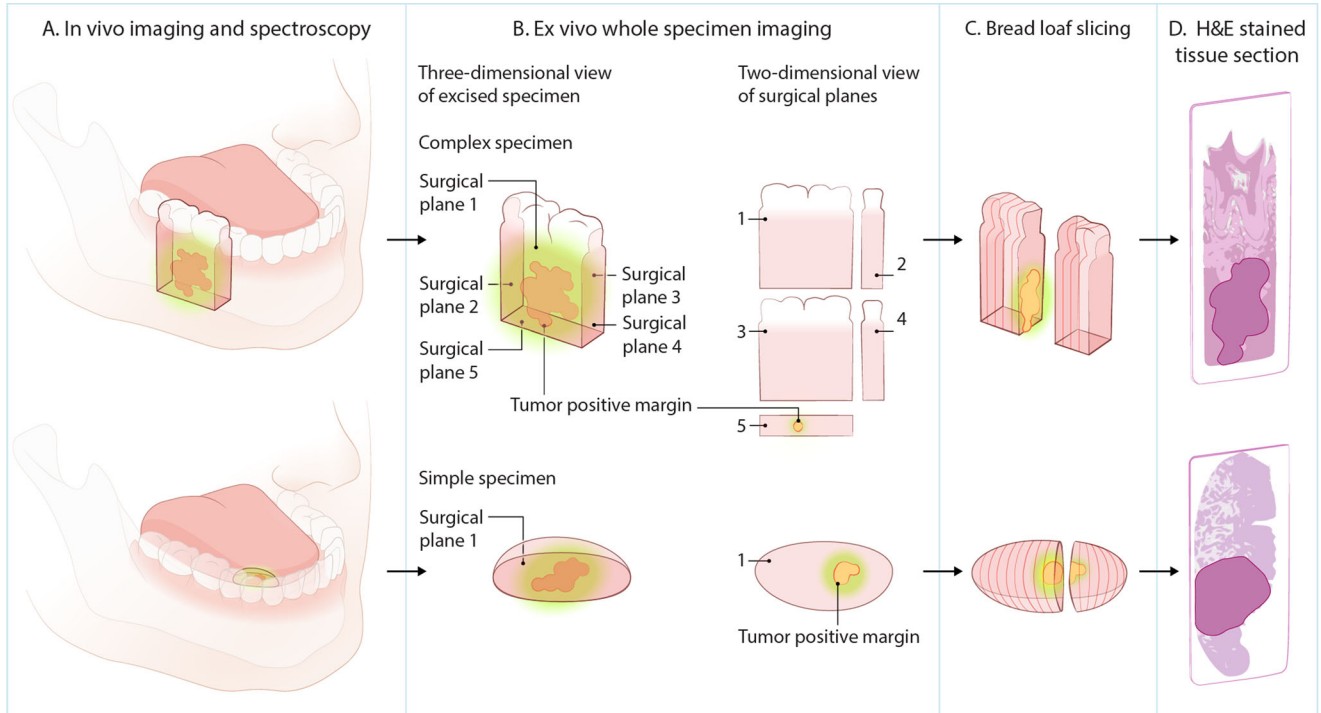

**Fig. 1 | Overview of study workflow. A** In vivo fluorescence imaging of the tumor. **B** Back table imaging of the excised specimen. Fluorescence imaging is performed from all surgical planes of the specimen. In the case of a complex specimen, multiple surgical planes can be identified and imaged, and in the case of a simple specimen, only one surgical plane per specimen is imaged. Fluorescent spots are observed in image 5 (top row) and image 1 (bottom row). **C** Bread loaf slicing of the specimen and fluorescence imaging of all bread loaf slices. **D** Correlation of the fluorescent spots relate to tumor-positive margins on histopathology.

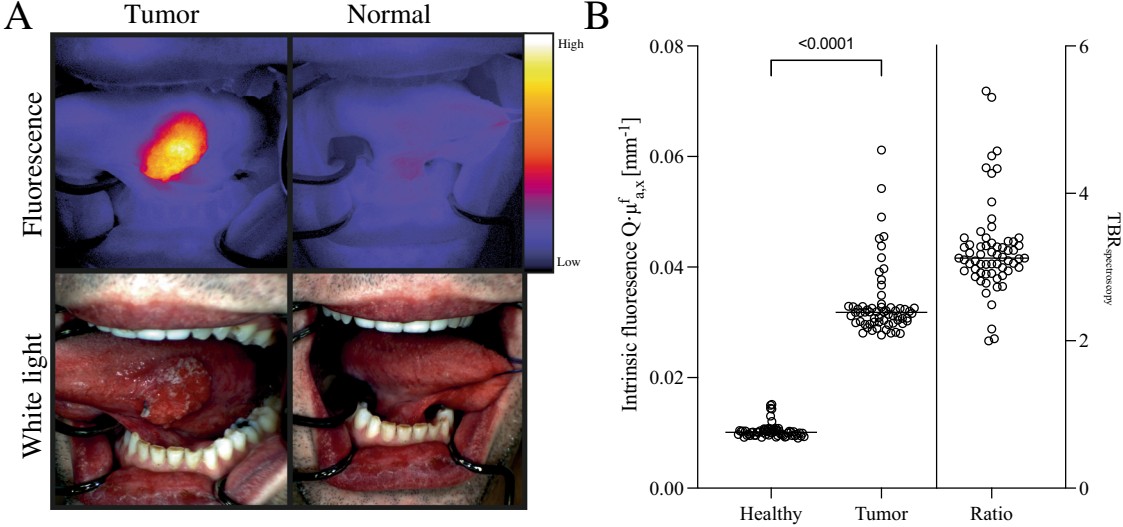

**Fig. 2 | In vivo imaging and spectroscopy results. A** In vivo fluorescence molecular imaging shows a sharp demarcation of a tumor on the lateral tongue. **B** In vivo multi-diameter single-fiber reflectance, single-fiber fluorescence contact measurements were performed in $n = 63$ tumors, showing significantly higher intrinsic fluorescence ($Q \cdot \mu^f_{a,x}$ [mm$^{-1}$]) in tumor (3.3 (2.7–6.1) × 10$^{-2}$ mm$^{-1}$) compared to normal tissue (1.0 (0.9–1.5) × 10$^{-2}$), one-sided $p = 0.0001$ using Wilcoxon signed rank test. Source data are provided as a Source Data file.

the diagnostic accuracy of the technique more precisely, we divided the close margins into two groups: 1–3 mm ($n = 20$) and 3–5 mm ($n = 17$). An SBR cut-off of ≥1.5 was optimal for both 1–3 mm and 3–5 mm close margins. We identified 1–3 mm margins with a sensitivity of 79.2% (95% CI 58.6–91.0%), a specificity of 76.1% (95% CI 58.6–87.7%), 52.7% (95% CI 42.0–74.7%) PPV, 91.5% (95% CI 74.6–97.5%) NPV, and an AUC of 0.78 (95% CI 0.67–0.89). We detected 3–5 mm margins with a sensitivity of 58.8% (95% CI 36.0–78.4%), a specificity of 76.1% (95% CI 58.6–87.6%), 37.0% (95% CI 20.7–57.0%) PPV, 88.5% (95% CI 72.5–95.7%)

NPV, and an AUC of 0.65 (95% CI 0.52–0.79). With an SBR of ≥1.5, we identified all 14/14 tumor-positive margins, 16/20 (80%) of close 1–3 mm margins, 10/17 (59%) of close 3–5 mm margins, and 17/71 (24%) of tumor-negative margins. Representative examples are shown in Fig. 5.

## Margin analysis on patient level

The 14 tumor-positive margins were detected in the surgical specimens of 13 patients with a sensitivity of 100% (95% CI 84.9–100%) on

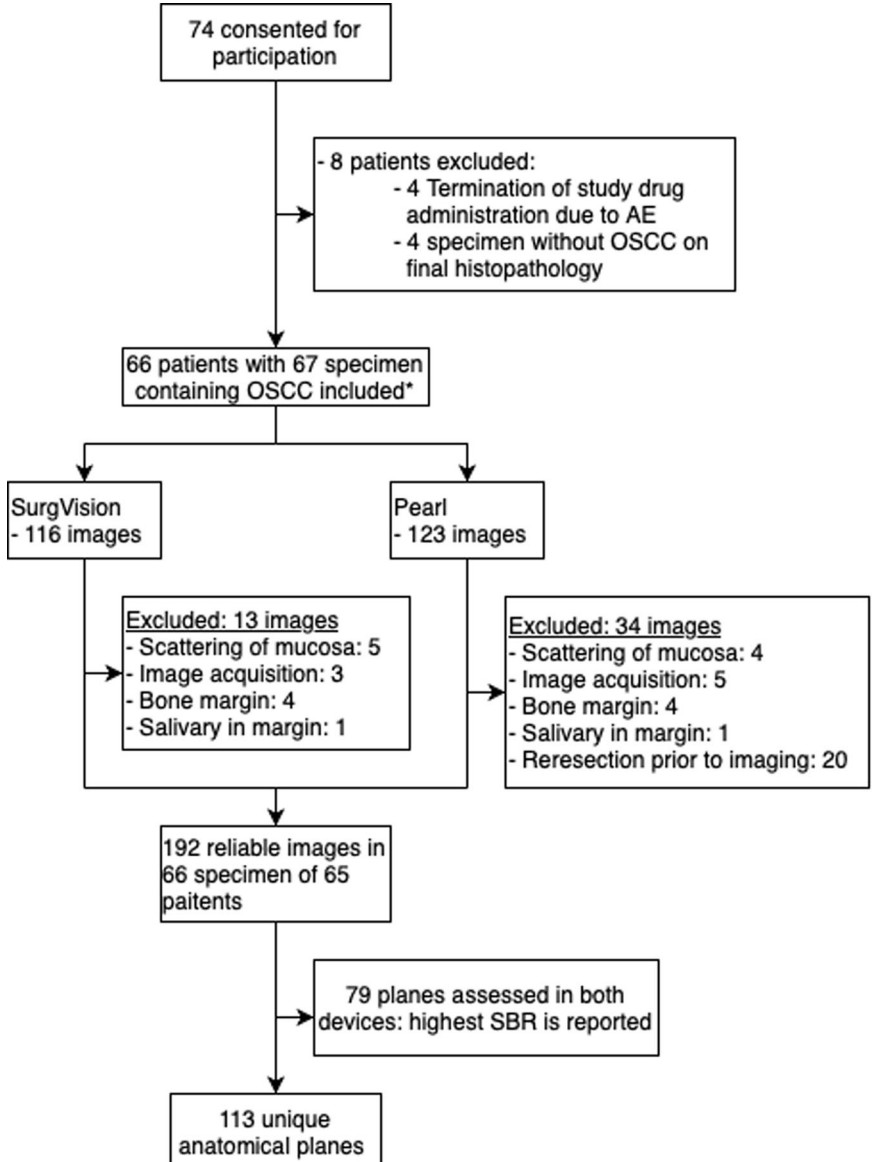

**Fig. 3 | Flowchart of patient inclusion and image acquisition.** OSCC oral squamous cell carcinoma, AE adverse event, SBR signal-to-background ratio. *One patient presented with two primary tumors.

the patient level (for specificity, see below). In 10 of these 13 patients, margin status was the independent indication for adjuvant chemoradiotherapy, and in three cases, it was also indicated based on tumor characteristics or lymph node involvement. A total of 37 close margins were observed in 34 patients on final histopathology; 26 of these close margins in 23 patients were detected with FMI with a sensitivity of 67.4% (95% CI 64.2–71.1%) on the patient level. In the latter group, radiotherapy was indicated solely on margin status in 20 patients. Finally, 17 margins in 14 patients showed false-positive resulting in a specificity of 41.7% (95% CI 40.0–43.0%) on the patient level, meaning unnecessary adjuvant resections would have been advised in these cases.

**False-negative and false-positive results**
Using an SBR of 1.5 (below an SBR of 1.5, the observers did not detect fluorescent spots) as a cut-off value, 11 false-negative close margins ($n = 4$ in the 1–3 mm margin width group and $n = 7$ in the 3–5 mm margin group) were found. Three out of four false-negatives of 1–3 mm showed small tumors with a low number of viable tumor cells combined with necrosis ($n = 1$) or extensive inflammation

encompassing the tumor ($n = 2$). For the remaining missed 3–5 mm margins ($n = 7$), immunohistochemistry could not explain the false-negative results. Representative examples are provided in Supplementary Fig. 2. Since most unexplained missed margins occur in the 3–5 mm margin group, we surmise these are due to limited depth information of the current SBR approach. We found 17 false-positive margins in 14 patients when using an SBR of ≥1.5. In 5/17 cases, salivary glands were localized in the deep margin; these are known to have EGFR expression[29] (Supplementary Fig. 3). Usually, the surgeon can determine the presence of salivary glands in the resection margin using visual and tactile information since the structure of salivary gland tissue has a distinct clinical appearance. In 2/17 cases, we observed a large artery located in a previously transplanted skin flap. Neither H&E histopathology nor EGFR immunohistochemistry could explain the increased fluorescent contrast in the remaining fluorescent false-positives. This increased contrast could have been the result of differences in tissue optical properties, differences in tissue geometry (i.e., angle of incidence), or nonspecific accumulation of the fluorescent tracer due to passive mechanisms (e.g., enhanced retention and permeability effect).

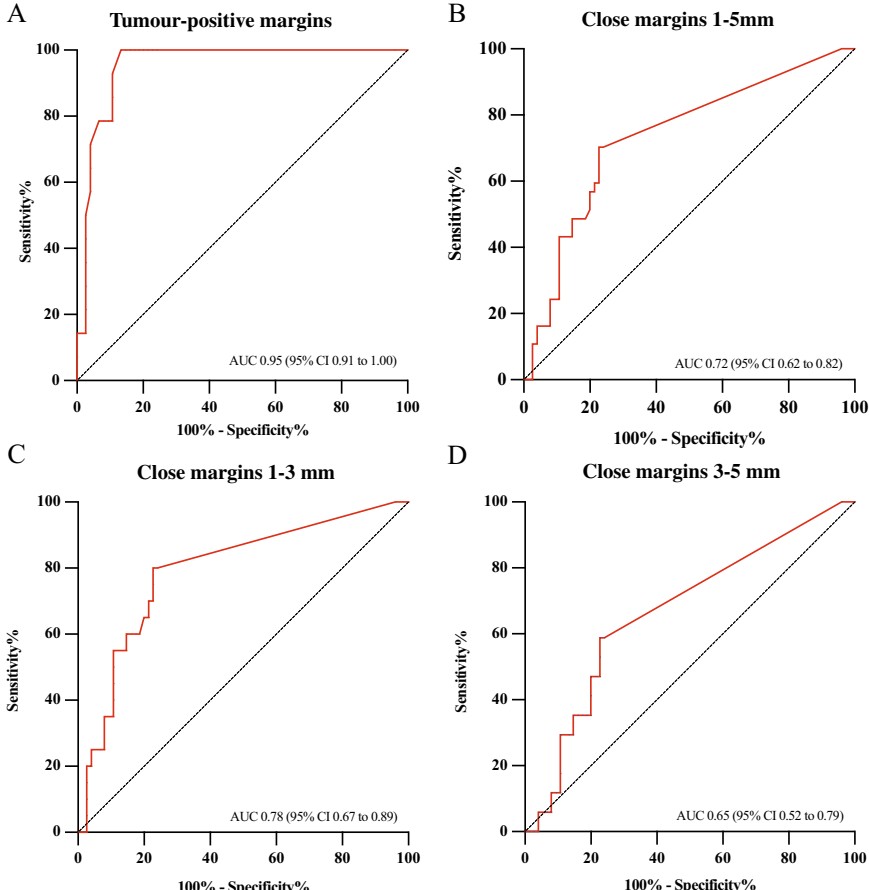

**Fig. 4 | Receiver operating characteristics of ex vivo margin assessment.**
Receiver operating characteristics for the detection of tumor-positive margins ($n = 14$, panel **A**), close margins ($n = 37$, panel **B**). To further determine the accuracy of our approach, we subdivided close margins in 1–3 mm ($n = 20$ panel **C**) and 3–5 mm ($n = 17$, panel **D**). Source data are provided as a Source Data file. AUC area under the curve, CI confidence interval.

Intrinsic fluorescence in previously irradiated tumors ($3.1$ $(2.8–6.1) \times 10^{-2}$ mm$^{-1}$) was not significantly different compared to non-irradiated tissue ($3.2$ $(2.8–5.4) \times 10^{-2}$ mm$^{-1}$) ($p = 0.85$, Supplementary Fig. 4). The presence of bone in the specimen (i.e., mandibulectomy) resulted in a significantly greater AUC for close margins ($z = 3.28$, $p = 0.005$) but not for tumor-positive margins ($z = 1.50$, $p = 0.067$) (Supplementary Fig. 5). In planes imaged by both the SurgVision Explorer Air® and the Pearl-Trilogy®, we observed a Pearson correlation for clustered data of 0.87. AUCs of positive margins were not different between the SurgVision Explorer Air® and the Pearl-Trilogy® (AUC of 0.92 (95% CI 0.79–1.0), 0.99 (95% CI 0.97–1.0), bootstrap $p = 0.11$, respectively), and neither for close margins (AUC of 0.75 (95% CI 0.64–0.86), 0.77 (95% CI 0.67–0.87), bootstrap $p = 0.48$, respectively). (Supplementary Fig. 6). To test imaging consistency, we performed manual segmentation of selected regions of repeated imaging of the fluorescence phantom using the SurgVision Explorer Air®. An average standard deviation of 10% was found (Supplementary Fig. 7).

### Safety data
In all patients enrolled in this study ($n = 74$), four (5%) adverse events were observed, which led to the termination of the study drug administration. All occurred during the administration of the predose of cetuximab. These included two serious adverse events (i.e., anaphylactic reaction with hypotension, CTCAE grade 3) and one grade I adverse event (rash, minimal angioedema). One grade I adverse event unrelated to the study drugs (vasovagal collapse) was also observed. None of these events caused a delay in the planned surgery. A complete overview of all adverse events (both related and unrelated) can be found in Supplementary Table 1.

## Discussion
Our ex vivo fluorescence molecular imaging approach using cetuximab-800CW during oral squamous cell carcinoma surgery not only detected tumor-positive margins with 100% sensitivity but also the majority of close margins. Intrinsic cetuximab-800CW fluorescence showed increased signal in all tumors compared to normal mucosa, with a median tumor to normal mucosa ratio of 3.1. FMI detected three malignant and premalignant satellite lesions that were missed by standard of care. In 10 out of 13 patients with tumor-positive margins, adjuvant chemoradiotherapy was solely based on margin status and intraoperative detection, and subsequent correction of the margins could have prevented this, had these margins been detected. In 20 (77%) patients with close margins, adjuvant radiotherapy could have been avoided through intraoperative correction of the margin. Consequently, the approach presented here—including the tracer-dose combination, interval and specimen-driven imaging performed directly after tumor excision—could be used immediately in intraoperative decision-making.

Dose-escalation studies have suggested that FMI has the potential to improve clinical outcomes in OSCC surgery[24,29–31]. Previously, we showed that pre-dosing with unlabeled cetuximab results in increased contrast between tumor and normal tissue[29]. This may be the result of preventing rapid plasma clearance of the tracer and occupying off-target receptors in normal tissue[27,28]. With our uniform dosing strategy, we obtained a consistently higher fluorescent signal in tumor tissue

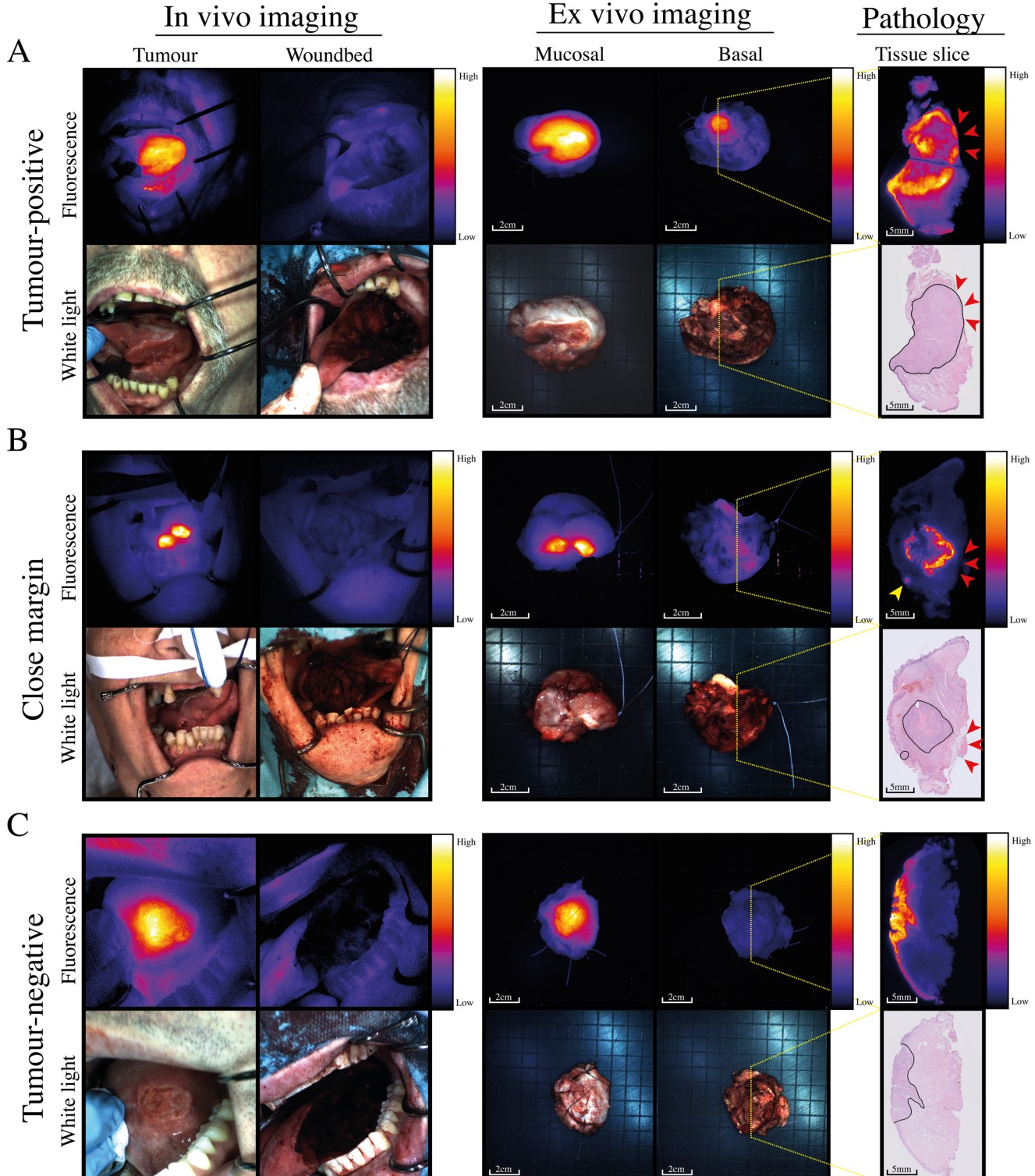

**Fig. 5 | Representative examples of a tumor-positive margin, close margin and tumor-negative margin.** Representative examples of **A** a tumor-positive margin, **B** a close margin, and **C** a tumor-negative margin. In vivo fluorescence imaging shows sharply demarcated tumors compared to adjacent tissue (upper left images), and after excision, no fluorescence can be detected in the wound bed (upper right images). On the tissue slices, the tumor is delineated with a solid black line. Panel (**A**) shows a fluorescent spot with an SBR of 4.0 on the excised specimen, corresponding to a tumor-positive margin (red arrows). Panel (**B**) shows a fluorescence spot with an SBR of 2.3, revealing a close margin of 2.2 mm (red arrows). The yellow arrow indicates a fluorescent lesion in the mucosa, which corresponds to the tumor spreading mucosally. In panel (**C**), no fluorescent signal is seen in the margin, corresponding to a tumor-negative margin. Tumor tissue is demarcated with a solid black line on tissue slides.

compared to the adjacent tissue, regardless of tumor location, previous surgery or radiotherapy in the oral cavity.

Van Keulen et al.[26] described a sentinel margin approach to identify the closest deep margin of a surgical specimen using the highest fluorescence peak. More recently, they showed in a group of 18 patients that this approach may be superior to the surgeon's judgment in identifying the closest margin[32]. We are building on this concept of assessing the surgical margin based on fluorescence peaks and have shown in the present study that this technique can reveal a tumor-positive margin prior to histopathological examination of the tissue.

An important aspect of using FMI for intraoperative margin assessment with an SBR approach is that it requires an experienced clinician or technician to execute the procedure and therefore requires training for assessors. Ultimately, standardized image interpretation protocols and training of assessors are advisable for easy implementation of this technique[33]. We found that, in the case of inadequate margins, the accuracy was inversely related to the width of the overlaying normal tissue margin. The accuracy was higher in overlying (inadequate) normal tissue margins between 1–3 mm compared to inadequate margins with thicker overlaying normal tissue of 3–5 mm. An optical technique that would allow for estimating the depth of origin of the fluorescence signal in the tissue could further improve accuracy. Such an optical device currently is not available, but recent developments in FMI camera systems have shown the potential to overcome this problem, such as angular restriction fluorescence optical projection tomography and spatial frequency domain imaging[34–36].

From the data of this study, we suggest using SBR cut-off values based on Youden's index that can be used for intraoperative margin assessment (i.e., SBR ≥ 2 and SBR ≥ 1.5). Although an SBR of ≥2 identifies all tumor-positive margins with high specificity, to improve close margin detection with FMI, we advocate using an SBR of ≥1.5 as cut-off. Since we found a limited PPV with an SBR of >1.5, we propose to perform fluorescence-guided fresh frozen sectioning from the excised specimen specifically aimed at the identified spot, which was not done in the current study to allow for an exact correlation of the margin width with the identified spots. This approach would enable the detection of all tumor-positive margins and the majority of close margins, potentially saving 10/13 patients from adjuvant chemoradiotherapy and 20/26 patients from adjuvant radiotherapy.

In conclusion, we have shown that EGFR-targeted fluorescence molecular imaging with cetuximab-800CW accurately identifies tumor-positive surgical margins of oral squamous cell carcinoma. This phase II FMI diagnostic accuracy study confirms the safety and potential of this technique and justifies a phase III multicentre study using the suggested strategy of the SBRs to determine the clinical impact in the treatment of OSCC.

## Methods
### Study design and participants
This prospective, single-arm, single-center phase II diagnostic accuracy study was performed at the University Medical Centre Groningen, the Netherlands. Patients eligible for inclusion were older than 18 years, had histologically confirmed OSCC, and were scheduled for surgical removal of the tumor. No pre-selection was made regarding the T-stage or sublocation of the tumor or prior treatment of the oral cavity with surgery or radiotherapy/chemoradiotherapy. Exclusion criteria are listed in Supplementary Information (Research study protocol, Section 4.3). The study protocol (available in the Supplementary Information file) was approved by the medical ethical review committee (METc) at the University Medical Centre Groningen (METc 2016/395) and was conducted according to the principles of the Declaration of Helsinki (adapted version Fortaleza, Brazil, 2013) and Good Clinical Practice. The trial was registered at www.clinicaltrials.gov (NCT03134846). Written and oral informed consent was obtained from all patients prior to any

study-related procedure. Specifically, the authors affirm that study participants provided informed consent for the publication of the images in Figs. 2 and 5 and Supplementary Fig. 1. Patients were enrolled between January 17, 2019, and November 29, 2021.

### Procedures
Clinical-grade cetuximab-800CW was produced at the Good Manufacturing Practice licensed facilities at the University Medical Centre Groningen. Briefly, commercially available cetuximab (Erbitux®) 5 mg/mL was conjugated to IRDye800CW NHS Ester (LI-COR Biosciences, Lincoln, NE, USA) under regulated conditions with a dye:antibody ratio of 2:1. The solution was purified using PD-10 buffer at 1 mg/ml exchange columns (GE Healthcare, Chicago, IL, USA). Cetuximab-800CW was formulated in a sodium-phosphate buffer at 1 mg/mL concentration and sterile filled into injection vials[37].

Patients received the study drugs intravenously, preceded by 2 mg clemastine according to standard of care cetuximab treatment. We administered 75 mg of unlabeled cetuximab to prevent rapid plasma clearance and occupy off-target receptors, followed by 15 mg cetuximab-800CW after 1 h[29]. Before and after the administration of the study drugs, vital signs were recorded. If no complications occurred, patients were discharged.

Two days after tracer administration, surgery was performed according to standard of care in our hospital. Routinely, CT and/or MRI were preoperatively available, on which the primary resection was planned. The tumor was removed with an estimated clinical margin of 1 cm. Fresh frozen sectioning of the margins was not performed routinely. Concomitant neck dissections or sentinel node biopsies were performed based on cTN classification, according to Dutch guidelines. In the case of clinical suspicion of an irradical resection intraoperatively, an extra resection was performed and attached to the primary specimen in the same setting. The tumor was removed with a margin of clinically uninvolved tissue, taken approximately 1 cm based on visual and tactile information to aim for a histopathological margin of >5 mm. The SurgVision Explorer Air® (SurgVision GmbH, Munich, Germany) was used for in vivo imaging and benchmarked using a fluorescence phantom prior to surgery[38]. We performed in vivo fluorescence imaging to visualize the primary tumor prior to incision with a set exposure time of 50 ms and gain of 100. Fluorescence molecular imaging was not used to outline the surgical margin in vivo. Multi-diameter single-fiber reflectance, single-fiber fluorescence (MDSFR/SFF) spectroscopy contact measurements were obtained to quantify intrinsic cetuximab-800CW tracer fluorescence by correcting the fluorescent signal for tissue optical properties (hereinafter referred to as 'intrinsic fluorescence')[15,20–23]. These measurements were obtained in triplicate from both tumor and normal mucosa, and median values are reported. After the excision of the tumor, we performed intraoperative fluorescence imaging of the intraoral wound bed and adjacent normal tissue. If a fluorescent spot was observed during in vivo imaging, a biopsy was taken of its location to determine its histopathology.

The freshly excised surgical specimen was imaged in what is referred to as the 'back table phase'. Ex vivo, specimen-driven imaging allows for controllable imaging parameters, resulting in consistent and reproducible images which were used for intraoperative margin analysis[8,15,39–41]. Two closed-field imaging systems were used in parallel: the SurgVision Explorer Air® coupled to a dedicated closed-field imaging box (Vault, SurgVision GmbH, Munich, Germany) and the PearlTrilogy® (LI-COR Biosciences, Lincoln, NE, USA).

Imaging was performed intraoperatively immediately after resection. For reliable fluorescence imaging results, the camera was placed perpendicular to the tissue of interest. All surgical planes containing a deep margin were imaged; therefore, the number of acquired images per specimen depended on the size and complexity of the specimen. For instance, a small tongue tumor specimen would result in one image containing a deep margin, and a more complex

resection of a maxillary carcinoma could result in five (Fig. 1). We excluded images in which reliable interpretation of fluorescence images was not feasible. Excluded images either had the surface of the specimen not perpendicular to the camera, a margin only consisting of bone, intraoperative extra resections performed and attached to the specimen prior to fluorescence imaging, or reflection of light in mucosal tissue interfered with the evaluation of the resection margin.

All images were scaled to the maximum fluorescence intensity observed in the tumor of each patient, as variations in tumor biology exist with each patient. A region of interest was drawn around each identified fluorescent spot on the excised specimen. A background region of interest was drawn on the same fluorescence image that included adjacent tissue of the same origin (e.g., connective tissue, muscle) without the fluorescent spot. A signal-to-background ratio (SBR) was calculated by dividing the mean fluorescence intensity of the spot by the mean fluorescence intensity of the background. When a fluorescent spot was identified with only one of the devices, the SBR of this fluorescent spot was used for analysis. In case a fluorescent spot was found using both imaging devices, the highest SBR (i.e., from either the Pearl-Trilogy® or the SurgVision Explorer Air®) was used for definitive analysis to make sure that no at-risk margins were missed. If no fluorescent spot was identified in the margin, the SBR was set at one. Fluorescence imaging and analysis were performed in approximately 5 min, depending on specimen size and complexity.

After completing the fluorescence imaging protocol, the surgical specimen was submitted to the Department of Pathology and formalin-fixed for at least 24 h according to the standard of care. The formalin-fixed specimen was inked for orientation purposes and serially sliced into 3–4 mm thick tissue slices. All tissue slices were imaged in the Pearl-Trilogy® to obtain cross-sectional fluorescence images of the tumor that allow for correlation with final histopathology. After paraffin embedding, a 4 μm section was cut from each tissue slice and stained with hematoxylin & eosin (H&E). A head and neck pathologist, blinded for fluorescence imaging results, delineated the tumor and determined the tumor margins in each 4 μm tissue section, according to the standard of care. The final margin status was classified as tumor-positive (<1 mm), close (1–5 mm) or tumor-negative (≥5 mm), as defined by the Royal College of Pathologists[42]. This process allows for an exact and constant determination of the margin width at the identified spots. If multiple spots or inadequate margins (i.e., margins <5 mm) were observed within one surgical plane of the specimen, all were included in the analysis.

### Endpoints

The trial's primary endpoints were the FMI detection rate of tumor-positive surgical margins and the cut-off value for SBR that can be used for intraoperative detection of tumor-positive margins. The secondary endpoints were the detection rates of close surgical margins, the in vivo fluorescence contrast between tumor and adjacent tissue as determined by MDSFR/SFF defined as $TBR_{spectroscopy}$, and the tolerability and safety of cetuximab-800CW, for which adverse events were graded according to Common Terminology Criteria for Adverse Events (CTCAE) version 5.0.

### Statistical analysis

Historical data in our center showed a tumor-positive margin rate of 15–20%. Consequently, in our study design, we included 70 patients and expected 14 to have a tumor-positive margin. Considering the EGFR overexpression rate of 90% in OSCC, potentially leading to inadequate fluorescence in 10% of the tumors, we expected to detect at least 12 out of 14 tumor-positive margins. Given the sample size, this would result in a sensitivity of 86% (95% CI 60–96%), yielding sufficient precision with regard to the expected impact on real-time intraoperative margin assessment, allowing the informed design of a

subsequent comparative randomized study. Estimates of specificity were expected to be even more precise given the predicted larger number of patients with tumor-negative margins. Descriptive statistics were performed on the patient demographics, clinical and pathological data. Data were presented as median + range, mean + standard deviation, frequencies, and percentages. Imaging data are presented as relative values (i.e., SBR). MDSFR/SFF values are reported as intrinsic fluorescence, defined as a product of the quantum efficiency across the emission spectrum, where Q is the fluorescence quantum yield of IRDye-800CW and $\mu_{af}$ [mm-1] is the tracer absorption coefficient at the excitation wavelength. All patients who completed study procedures and showed tumor on histopathology were included in the analyses. Receiving operating characteristic (ROC) curves were plotted to determine the discriminative ability of intraoperative FMI for inadequate margins. No post-processing (i.e., binning or smoothing) was used. Area under the ROC curves (AUCs) were estimated together with 95% confidence intervals while considering potential clustering of data within patients[43]. The optimal cut-offs for detecting tumor-positive and close margins were determined using Youden's index. We estimated margin-level sensitivity, specificity, and positive and negative predicted values and corresponding 95% confidence intervals using an ANOVA-type Wilson score estimation that considers clustering within patients[44]. To compare clustered AUCs between bone-involved margins and margins containing soft tissue, we used z-tests. Furthermore, we compared margin-level SN values between measurements from both imaging devices (the Pearl-Trilogy® and the SurgVision Explorer Air®) using a Pearson correlation coefficient for clustered data[45] and compared the resulting clustered AUCs for margin status between the two techniques using a 2000-fold cluster bootstrap. A two-tailed p-value <0.05 was considered significant. GraphPad Prism (version 8.0, GraphPad Software Inc, San Diego, California, USA) and R (version 4.2.2. for MacOS, R Foundation for Statistical Computing, Vienna, Austria) were used for statistical analysis and graph design. ImageJ Fiji (Version 2.3.0/1.53f) was used for fluorescence image analysis.

### Reporting summary

Further information on research design is available in the Nature Portfolio Reporting Summary linked to this article.

## Data availability

All imaging data, safety data, clinical details and laboratory data (i.e., restricted to non-identifying data) are available from the corresponding author on request. Data can be inquired by the corresponding author (M.J.H.W., m.j.h.witjes@umcg.nl). The data will be saved for a minimum of 20 years, in concordance with the Dutch legislation. Upon request, data can be made available to third parties for up to six weeks. The study protocol is available as Supplementary Note in the Supplementary Information file. The remaining data are available within the Article, Supplementary Information or Source Data file. Source data are provided with this paper.

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

## Acknowledgements

We thank all patients who participated in this study. We would also like to thank Karien Kreeft-Polman, Ellen van den Ende-Schaap, and Hilde

Bouma-Boomsma for their help in recruiting patients and Maaike Barentsen, Erik Bleuel, and Lilo Janssens for their assistance in specimen processing. Finally, we thank the Dutch National Cancer Society (KWF, Koningin Wilhelmina Fonds) for their financial support (grant number RUG 2015-8084) granted to M.J.H.W. In this investigator-initiated trial, the sponsor has had no influence on the study design, data collection or analysis and writing of the manuscript.

## Author contributions

All authors have critically reviewed and approved the final version of the manuscript, and all authors fulfill the Committee on Publication Ethics requirements for authorship. Specifically, M.J.H.W. and G.M.v.D. designed the study. J.G.d.W., J.V. and F.J.V. performed data acquisition, analyzed and interpreted data and drafted the manuscript. D.R. performed spectroscopy data analysis. W.T.R.H. and M.D.L. were involved in the production of the study drug (cetuximab-800CW). B.v.d.V. and J.J.D. were involved in histopathological analyses. M.J.H.W., K.P.S., S.A.H.J.d.V., B.E.C.P. and G.B.H. performed surgical procedures. S.G.E. performed statistical analyses. E.L.R. and W.B.N. reviewed the manuscript. M.J.H.W. supervised the study, interpreted data and supervised the writing of the manuscript.

## Competing interests

G.M.v.D. is CEO of Tracer Europe B.V./AxelaRx. B.v.d.V. is a member of the Scientific Advisory Board of Visiopharm, for which compensation is received by the University Medical Centre Groningen. The remaining authors declare no competing interests.
