## [Peer Review File · Nature Communications]

EGFR-targeted fluorescence molecular imaging for intraoperative margin assessment in oral cancer patients: a phase II trialREVIEWER COMMENTS

Reviewer #1 (Remarks to the Author): with expertise in oral cancer, clinical, surgery

The biggest challenge in surgical resection of oral cancer is the determination of the deep margin, which is often done by estimating depth of invasion by palpation and correlating this with imaging findings. As a result, this is often the margin that is returned as close of positive (although not as often as the citations in the introduction of 40%, but closer to the power calculations of 20%). However, it has also been a question of whether tumors where margins are positive or close are ascribed to a different tumor biology (ie more infiltrative and hence less 'palpable'.

In this manuscript, the authors have taken on the challenge of deep margin assessment using EGFR-based fluorescent markers as part of a phase 2 trial, recruiting and analysing 66 tumors as a result, and this is a commendable exercise. While the choice of markers is always one that is debatable, the ability to modify a well-tolerated drug such as cetuximab, allows for the easy adoption of this technique. What is less known here is the imaging techniques/technologies used for and perhaps some explanation of the two different imaging techniques would be useful in the manuscript. This is a definitely a challenging topic to address and execute and I commend the authors for conducting this trial in a robust manner. I have a few suggestions on improvements to the study and manuscript:

1. I feel that the figure describing their techniques has been buried in Figure 5, when this is better shown top front. Perhaps they should include the whole study including in vivo and in vitro imaging for clarity, and provide numbers in this figure.
2. Also missing is the normal practice of the surgeons who performed these procedures on how they determined tumor extent and planned their margins for excision (distances, assessments, intraoperative change in plans etc).
3. The abstract should emphasize the AUC's in the ROC which was omitted
4. Were the tumors tested for EGFR expression, and did this affect the tumor margin assessment (false positive/negative)
5. were neck dissections routinely done and if so, why was that no included in the decision making process for RT or CRT.
6. It has been over a year since this study and hence I would like to see some survival stats, and ask if re-resection/aggressive treatment of surgical margins altered outcomes
7. The methods section did not suggest that any fluorescent imaging was performed during pathology, yet Figure 4 shows two images to suggest that. Can the authors explain this omission/discrepancy. If you did take parallel path/H&E and fluorescent images, did you perform image analysis superimposing these two to demonstrate concordance to the cellular level?

Reviewer #2 (Remarks to the Author): with expertise in image-guided surgery, clinical trials

Fluorescence molecular imaging for intraoperative margin assessment in oral cancer patients: a prospective, phase II imaging trial

The manuscript by Wit et al, details a Phase II fluorescence imaging trial to evaluate its utility for assessment of margins during oral cancer surgeries. The team used IRDye-800CW conjugated to Cetuximab as the tumor-specific contrast agent given that EGFR overexpression is common in oral squamous cell carcinoma. The team evaluated overall margin positivity as well as the depth of the positive cells within the tumors. Overall this is a good study with well-defined outcomes this is important to the field of fluorescence guided surgery. However, some additional details and clarifications are needed in order to provide sufficient context to evaluate the significance of the work, detailed as follows.

1. SBR is generally defined in the field as signal-to-background ratio rather than spot-to-background. Strongly suggest changing to signal-to-background ratio to be consistent with the terminology used in

the field.

2. It isn't clear why cetuximab was selected for EGFR targeting vs. panitumumab since panitumumab is known to have an improved safety profile. Indeed adverse events upon infusion with cetuximab were seen. Please provide detail on this antibody choice for the study.
3. The pre-dosing strategy with unlabeled antibody and its utility should be covered in the introduction section.
4. Why was the SBR cut off set at >1.5 for close margins, but at >2 for tumor-positive margin detection overall? This is not explained.
5. The standard of care for an R0 resection needs to be defined in the introduction for contextual understanding of the study parameters. It isn't mentioned in the current manuscript until the end of the methods section.
6. What system was used to image in vivo in patients? It appears based on the methods this system may be the SurgVision Explorer Air, but it isn't clear. Please provide additional detail.
7. Would this methodology always be used for "back table" imaging to quantify tumor margins or would it also be used for imaging the tumor within the patient quantitatively in its future clinical implementation?
8. Why is the sensitivity and specificity of the Pear-Trilogy generally better than the SurgVision Explorer Air system?
9. The caption on Supplementary Fig 3 isn't clear. Specifically, it is stated that "The detection rate for tumor-positive margins is 100% for specimens with (Panel A) and without bone (Panel B)." While the AUC of the ROC in Panel A appears to be 1.0, this isn't the case for Panel B. Please clarify.

Reviewer #3 (Remarks to the Author): with expertise in fluorescence-guided surgery

The manuscript presented by de Wit et al, summarizes the clinical findings from a prospective, Phase II fluorescence molecular imaging trial of oral squamous cell carcinoma patients. The diagnostic accuracy of cetuximab-800CW to detect oral cancers was investigated 2 days after a 75 mg infusion of unlabelled cetuximab and 15 mg of cetuximab-800CW. Tumors were imaged in situ with the SurgVision Explorer Air and spectra were acquired with multi-diameter single-fibre reflectance, single fibre fluorescence spectroscopy to quantify cetuximab-800CW fluorescence. Ex vivo specimen imaged with the SurgVision Explorer Air and Pearl-Trilogy. Diagnostic assessment of the deep margin was performed in the operating room using spot-to-background ratios (SBR). The study resulted in high sensitivity (100%) and specificity (86%) for tumor-positive margins with an SBR equal to or greater than 2, and lower sensitivity (70%) and specificity (76%) for close margins using an SBR equal to or greater than 1.5. This work further validates the use of sentinel margin assessment to identify positive, or close, tumor margins in the surgical suite in real time. Overall, the study is well developed, and the results presented are strongly in favor of adding fluorescence molecular imaging to surgical resection of oral squamous cell carcinomas. Furthermore, the results provide quantifiable measures to guide the margin assessment that may result in significant improvement in patient outcomes. The clinical impact of this study and resulting Phase III trials could be significant in the reduction of adjuvant therapies and treatments. However, there are some clarifications that need to be addressed in terms of the image exclusion criteria, the ex vivo diagnostic accuracy results from the two imaging systems, and multivariate analysis of data (multiple images from the same resection).

Major comments:

1. The positive results of the study are well documented and explained. However, as the team is moving towards Phase III clinical testing, the more scientifically interesting question, and potential to improve upon the current results in continued testing, is to investigate why some specimen were miscategorized. This is only briefly mentioned in lines 159-172 of the Results, but no data is shown.
 - a. There were 11 false negatives, with n=4 in the 1-3 mm margin and n=7 in the 3-5 mm margin. Only the 1-3 mm margin were explained (low number of cells with necrosis or inflammation). What about the 7 miscategorized 3-5 mm margin? Was this size dependent? EGFR expression? Can the region be correlated back to the specimen and the actual TBR be calculated?
 - b. In the false-positive margins, there was n=5 patients that had salivary gland in the margin. However, in Fig. 2 salivary gland is listed as an exclusion criterion for 1 specimen. It is unclear how salivary gland is being treated, especially since the authors state that “the surgeon can determine the presence of salivary glands in the resection margin using visual and tactile information.” Were these salivary glands detected by the surgeon? Why were some excluded and others not? What if the salivary glands were involved?
2. Lines 301-306, Figure 2: Image exclusion criteria are not well defined in the procedures. The list of image exclusion criteria in Figure 2 do not match that in the procedures, lines 301-306. Salivary gland is excluded, as well as “scattering of mucosa”.
 - a. Can “scattering of mucosa” be more well defined in the procedure? Why was this an issue in 5 patients in the SurgVision system and only 4 patients in the Pearl system?
 - b. Please explain the salivary gland exclusion and why others were included.
3. The spot-to-background (SBR) results appear to be a combination of images from the Pearl and the SurgVision systems; however, it is unclear what combination of these two systems are used.
 - a. The SurgVision was better at determining the close margins than the Pearl system for both the SBR greater or equal to 1 and 1.5, however the results reported in Table 1, the abstract, and throughout the manuscript report higher values than each system alone. Please clarify since the combined results appear to be quite a bit higher.
 - b. My instinct is that one system is better at determining the close margins in the 3-5 mm range than the other. Is there correlation between SBR and depth for each system?
 - c. Is it clinically viable in the future to image every excised specimen with two systems? The Pearl is advantageous for positive margins, while the SurgVision is advantageous for close margins. Please discuss and provide analysis.
4. How were the statistics handled for multiple images from the same patient? Is this considered when determining the diagnostic accuracy? The descriptions of the statistical analysis in the procedures are vague.
5. All fluorescence images would benefit from being scaled to the same level, instead of the color map being scaled from “high” to “low”. Are they all scaled with the same “high” and “low” value? Or individually scaled.
6. How were multiple regions in the margin addressed? In lines 326-327, it is stated that if multiple regions in the margin were detected then all were included. However in Figure 4B, there is a close margin indicated on the larger circled tumor but there appears to be a satellite tumor region that is not addressed in terms of whether it is also a close margin. The fluorescence from second smaller region seems negligible. Addition of the measured SBR for these regions on the images would greatly help the reader interpret the results.

Other comments:

1. Lines 43-44 – Is the sentence starting “Fluorescence molecular imaging (FMI) has been explored...” only talking about oral squamous cell carcinomas? It could be argued that FMI is used for

margin/residual tumor in ovarian and lung cancer. This sentence just needs clarification that it is not globally speaking to all FMI.

2. What are the main differences between Supplementary Table 1 and Supplementary Table 2?
 - a. Supplementary Table 1 is never called out in the text of the manuscript and only some of the values are reported in the abstract and in the text.
 - b. It seems Supplementary Table 1 is the values determined as described in lines 311-317 (the highest SBR of the two systems)?
 - c. Supplementary Table 1 summarizes the results for close margin 1-3 and 3-5 but not 1-5, which is the value reported in the abstract and in lines 135-137. These data should be included in Table 1
 - d. The differentiation of close margins at 1-3 mm depth and 3-5 mm depth should also be included in Table 2.
 - e. Please clarify how many images were used from each system for the Table 1.
 - f. Please include the image numbers for all tables, as they varied between systems.
3. The language is confusing around the discussion of fluorescence ratios. The only ratio that is defined is the spot-to-background ratio (SBR) the following are also discussed:
 - a. Tumor to normal mucosa ratio (line 199)
 - b. Median ratio (line 115)
 - c. TBR (Figure 1B, second y-axis, line 489)
4. Figure 5 would benefit from a description on why the top figure of the figure is different than the bottom of the figure. Is it just location and number of surgical margins? Does this influence the number of images from the patient?
5. Was the phantom imaged on both the SurgVision and the Pearl systems? This is not stated but it appears that results from only one system are provided.
6. Lines 208-210 – The statement “Previously, we showed that predosing with unlabelled cetuximab can block off-target receptors, thus increasing tumor uptake of the fluorescent tracer and reducing the background signal.” is a very strong statement to make and is not adequately demonstrated in Reference 27. The data in Reference 27 shows that the fluorescence normal surrounding tissue stays relatively constant in most doses (excluding the 50 mg cetuximab-800CW) in Figure 4B and 4C. The additional images provided in Figure S6 do not adequately demonstrate significant off-target signal reduction due to receptor saturation. You do not histologically demonstrate that the signal outside of the tumor ROIs is actually from EGFR in normal tissue and you are providing representative data, not entire cohorts analyzed.

Reviewer #4 (Remarks to the Author): with expertise in biostatistics, clinical trial study design

This review pertains only to the statistical aspect of this manuscript.

The authors identified several cut-off values in spot-to-background ratio (SBR) for predicting several binary endpoints. Sensitivity, specificity, positive and negative predictive values, as well as the receiving operating characteristic (ROC) curves as well as area under ROC curves (AUC) are provided for each binary endpoint. The optimal cut-off values are determined by Youden's Index (i.e., maximum sensitivity + specificity). Understandably, most of the numbers and curves (Figure 3, B-D) are not ideal due to the limited sample size.

The reviewer would suggest performing a recursive partitioning regression model, in addition to the existing statistical methods. This is only a suggestion, not a requirement.

Point-by-point Response

"Fluorescence molecular imaging for intraoperative margin assessment in oral cancer 2 patients: a prospective, phase II imaging trial"

Jaron G. de Wit, Jasper Vonk, Floris J. Voskuil, Sebastiaan A.H.J. de Visscher, Kees-Pieter Schepman, Wouter T.R. Hooghiemstra, Matthijs D. Linssen, Sjoerd G. Elias, Gyorgy B. Halmos, Boudewijn E.C. Plaat, Jan J. Doff, Eben L. Rosenthal, Dominic Robinson, Bert van der Vegt, Wouter B. Nagengast, Gooitzen M. van Dam, Max J.H. Witjes

We would like to thank the 4 reviewers for thoroughly reading our submitted manuscript and providing us with constructive comments to improve our manuscript. To facilitate matching our responses and how we addressed reviewer's comments specifically, here we provide a detailed point-by-point response.

The reviewer's comments are stated in *italic*. Our responses are written in **blue font color**. Cited paragraphs from the revised manuscript are displayed in **brown font color**. New or rewritten sections in these sentences are underlined. In our updated manuscript, we highlighted amended sections with yellow background and additional sections with green background.

Reviewer #1 (Remarks to the Author): with expertise in oral cancer, clinical, surgery

The biggest challenge in surgical resection of oral cancer is the determination of the deep margin, which is often done by estimating depth of invasion by palpation and correlating this with imaging findings. As a result, this is often the margin that is returned as close of positive (although not as often as the citations in the introduction of 40%, but closer to the power calculations of 20%). However, it has also been a question of whether tumors where margins are positive or close are ascribed to a different tumor biology (ie more infiltrative and hence less 'palpable').

In this manuscript, the authors have taken on the challenge of deep margin assessment using EGFR-based fluorescent markers as part of a phase 2 trial, recruiting and analysing 66 tumors as a result, and this is a commendable exercise. While the choice of markers is always one that is debatable, the ability to modify a well-tolerated drug such as cetuximab, allows for the easy adoption of this technique. What is less known here is the imaging techniques/technologies used for and perhaps some explanation of the two different imaging techniques would be useful in the manuscript.

This is definitely a challenging topic to address and execute and I commend the authors for conducting this trial in a robust manner. I have a few suggestions on improvements to the study and manuscript:

1. I feel that the figure describing their techniques has been buried in Figure 5, when this is better shown top front. Perhaps they should include the whole study including in vivo and in vitro imaging for clarity, and provide numbers in this figure.

We agree with the reviewer that Figure 5, describing the study workflow, is better suited to show top front. We have renamed the figure as 'Figure 1' and have moved it accordingly. To better show the complete study workflow, we have changed the heading of panel A.

We now refer to Figure 1 in the first section of the Results:

(Lines 112-113) “The study workflow of imaging procedures, pathology processing and final histopathology is depicted in Fig. 1.”

2. Also missing is the normal practice of the surgeons who performed these procedures on how they determined tumor extent and planned their margins for excision (distances, assessments, intraoperative change in plans etc).

Surgical excision was performed according to the standard of care and therefore not extensively described in the manuscript. For clarity we have added the following sections to the Methods section:

(Lines 295-298) “Two days after tracer administration, surgery was performed according to standard of care. Routinely, CT and MRI were preoperatively available. The tumor was removed with a margin of clinically uninvolved tissue, taken approximately 1 cm based on visual and tactile information to aim for a histopathological margin of >5 mm.

(Line 302) “Fluorescence molecular imaging was not used to outline the surgical margin *in vivo*.”

3. The abstract should emphasize the AUC's in the ROC which was omitted

We thank the reviewer for this suggestion. We have added the AUCs for the detection of tumor-positive and close margins in the Results section of the abstract. The section now reads as follows:

(Lines 46-50) “In 65 patients with 66 OSCCs, an SBR cut-off of ≥ 2 detected tumor-positive margins with 100% sensitivity, 86% specificity, 58% positive predictive value (PPV), 100% negative predictive value (NPV), and an area under the curve (AUC) of 0.95. Close margins (1-5 mm) were detected with 43% sensitivity. An SBR of ≥ 1.5 identified close margins with 70% 52 sensitivity, 76% specificity, 60% PPV, 83% NPV and an AUC of 0.72.”

4. Were the tumors tested for EGFR expression, and did this affect the tumor margin assessment (false positive/negative)

EGFR immunohistochemistry was performed in our phase I fluorescence molecular imaging trial in OSCC patients (doi:10.7150/thno.43227), where we found that 96% of the tumors showed EGFR expression. Therefore, in the current manuscript we did not perform EGFR immunohistochemistry on all tumors. We only performed EGFR immunohistochemistry to study the false negative and false positive results.

Fluorescence false negative results showed EGFR expression in the tumor but showed limited numbers of viable tumor cells. For this trial we did not preselect on the size of the tumor. In the false positive results, we could not find EGFR expression in the adjacent tissue explaining the fluorescent signal.

(Lines 171-182): “Using an SBR of 1.5 (below an SBR of 1.5 the observers did not detect fluorescent spots) as a cut-off value, 11 false-negative close margins (n=4 in the 1-3 mm margin width group and n=7 in the 3-5 mm margin group) were found. Three out of four false negatives of 1-3 mm showed small tumors with a low number of viable tumor cells combined with necrosis (n=1) or extensive inflammation encompassing the tumor (n=2). We found 17 false-positive margins in 14 patients when using an SBR of ≥ 1.5 . In 5/17 cases, salivary glands were localized in the deep margin; these are known

to have EGFR expression²⁷. Usually, the surgeon can determine the presence of salivary glands in the resection margin using visual and tactile information, since the structure of salivary gland tissue has a distinct clinical appearance. In 2/17 cases, we observed a large artery located in a previously transplanted skin flap. Neither H&E histopathology nor EGFR immunohistochemistry could explain the increased fluorescent contrast in the remaining fluorescent false-positives. This increased contrast could have been the result of differences in tissue optical properties, differences in tissue geometry (i.e., angle of incidence), or nonspecific accumulation of the fluorescent tracer due to passive mechanisms (e.g., enhanced retention and permeability effect)."

Below we have included representative examples of false-negative and false-positive results. We prefer not to add these figures to the manuscript to keep the paper more concise. We can add the figures to the supplementary materials upon request by the reviewer or editor.

The figure above shows typical examples of false negative results. A) A false negative fluorescence result corresponding to a close margin of 1.9mm. Note that on the H&E slice, the tumor consists mostly of a large necrotic core with limited viable tumor cells, thus little EGFR receptors. This explains the absence of fluorescence in the centre of the tumor that results in insufficient fluorescent signal to create a spot in the resection margin. B) A false negative margin of 2.2mm, for which immunohistochemistry cannot provide an explanation for the absence of fluorescence signal in the margin.

The figure above depicts representative examples of false positive results. A) A false positive result due to salivary gland tissue in the margin, where we also observe fluorescence signal in the salivary gland tissue. B) A false positive fluorescent lesion in the margin, which cannot be explained by histopathology.

5. Were neck dissections routinely done and if so, why was that not included in the decision making process for RT or CRT.

The reviewer is correct that neck dissections or sentinel node biopsies were routinely performed according to the Dutch head and neck cancer guidelines. This was included in the decision-making process for RT or CRT, according to standard of care. We have described whether the margin status or other factors (e.g. lymph node involvement) formed the indication for adjuvant therapy in the Results section:

(Lines 158-159) “and in three cases it was also indicated based on tumor characteristics or lymph node involvement.”

We have added a clarifying statement to the Discussion section:

(Line 211-214) “In 10 out of 13 patients with tumor-positive margins, adjuvant chemoradiotherapy was solely based on margin status and intraoperative detection and subsequent correction could have prevented this, had these margins been detected.”

In our institute, we have poor experience with detecting intraoperative positive margins solely based on frozen sections, and therefore these are not routinely performed. However, with the data from this study we are confident that we will return to performing intraoperative frozen sections of fluorescent spots.

6. It has been over a year since this study and hence I would like to see some survival stats, and ask if re-resection/aggressive treatment of surgical margins altered outcomes

We agree with the reviewer that the ultimate goal of this technique is to improve the patient's prognosis. Yet, we have chosen not to present these data in the current manuscript as the primary outcomes of this study do not include recurrence or survival data. This would be confusing to the reader. Also, we think that the outcome of this study shows that it would be sensible to use the fluorescence of the margins as a tool to indicate where to take frozen sections, rather than based solely on the intensity of a fluorescent spot. Next to the fact that the study was not powered to provide reliable survival data. We have provided the requested data below. If the editor would find these data valuable, we could provide these data in the manuscript or supplementary data.

The data show that:

In the first 15 patients of this phase-II trial, we observed that patients with a fluorescent spot with a TBR/SBR >2 showed a tumor-positive margin in that area (n=5). Considering the importance of obtaining a clear margin, we discussed this observation with the Medical Ethics Committee of our center. Consequently, we were allowed to perform re-resections in case a fluorescent spot was observed in the deep margin.

In the following 50 patients, we advised the surgeon to reassess the resection margin in this specific region to possibly extend the surgery in all planes of which we observed a spot with a SBR >1.5 using the SurgVision Open Air® (which was available in the operating room). We analysed a total of 85 surgical planes in these 50 patients using this approach. 36 surgical planes contained a spot with SBR >1.5. These included 10 positive, 14 close and 12 negative margins. In 15/36 cases, an additional resection was performed. In the remaining cases, an additional resection was omitted due to lack of clinical suspicion (n=11) or the technical impossibility to excise additional tissue since it would lead to significant morbidity (n=10).

The 15 re-resections included 4 positive, 3 close and 8 negative margins. The cases in which a re-resection was omitted based on a lack of clinical suspicion, we found 3 positive, 5 close, and 3 negative margins. Using FGS, in 15/26 (58%) an intraoperative correction of the margins was correctly advised in cases where extension of surgery was possible. The 49 margins that were not considered at risk based on fluorescence imaging (i.e. SBR < 1.5), included 2 positive margins and 5 close margins. The two missed tumor-positive margins were later identified during postoperative analysis of the freshly excised specimen using the Pearl Trilogy imaging system.

In 63 patients (two patients participated twice), we report a median follow up of 2.96 (0.61 to 4.07) years. Twenty-three patients reported with local recurrence (n=9), secondary primary tumors (n=9) or late lymph node metastasis (n=6) occurring within 0,91 (0,04 – 2,45) years. Considering the differences in follow-up time, overall disease-free survival (DFS) was 1,92 (0,04 to 4,03) years. Looking at margin status and survival, DFS was 2,57 (0,48 to 4,03) years for tumor free margins, 1,95 (0,14 to 3,80) for close margins and 0,85 (0,04 to 3,74) for tumor positive margins.

Eighteen patients have deceased, of which 14 due to oral squamous cell carcinoma. Of these 14 patients, 8 had a tumor-positive margin, 4 a close margin and 2 a tumor-free margin during the initial surgery.

7. The methods section did not suggest that any fluorescent imaging was performed during pathology, yet Figure 4 shows two images to suggest that. Can the authors explain this omission/discrepancy. If you did take parallel path/H&E and fluorescent images, did you perform image analysis superimposing these two to demonstrate concordance to the cellular level?

The reviewer is correct that we did not clearly explain that fluorescence molecular imaging was performed during pathology processing. For clarity, the complete surgical specimen was imaged directly after surgery and subsequently during tissue processing at the Department of Pathology. The latter includes imaging of the formalin-fixed surgical specimen as well as the bread-loaf slices. We did not superimpose the fluorescent images on the corresponding H&E tissue section since this is only performed when it is needed to confirm the location of the tracer in the tumor tissue. We refer to our phase I study where cetuximab-800CW binding specificity was studied (doi:10.7150/thno.43227). In that study, we showed that the fluorescence signal co-localizes with tumor tissue. For the current study the relevant question was to identify fluorescent spots in the complete surgical specimen and assess what the outcome would be in terms of margin distance at the fluorescent spot. As such, the localizing capacity in the tumor is intrinsically tested as well.

We have added the following to the Methods section:

(Lines 344-345) "After completing the fluorescence imaging protocol, the surgical specimen was submitted to the Department of Pathology and formalin-fixed for at least 24 hours according to standard of care. The formalin-fixed specimen was inked for orientation purposes and serially sliced in 3-4 mm thick slices. All tissue slices were imaged in the Pearl Trilogy® to obtain cross-sectional fluorescence images of the tumor that allow for correlation with final histopathology. After paraffin embedding, a 4 µm section was cut from each tissue slice and stained with haematoxylin & eosin (H&E).

Reviewer #2 (Remarks to the Author): with expertise in image-guided surgery, clinical trials

Fluorescence molecular imaging for intraoperative margin assessment in oral cancer patients: a prospective, phase II imaging trial

The manuscript by Wit et al., details a Phase II fluorescence imaging trial to evaluate its utility for assessment of margins during oral cancer surgeries. The team used IRDye-800CW conjugated to Cetuximab as the tumor-specific contrast agent given that EGFR overexpression is common in oral squamous cell carcinoma. The team evaluated overall margin positivity as well as the depth of the positive cells within the tumors. Overall this is a good study with well-defined outcomes this is important to the field of fluorescence guided surgery. However, some additional details and clarifications are needed in order to provide sufficient context to evaluate the significance of the work, detailed as follows.

1. SBR is generally defined in the field as signal-to-background ratio rather than spot-to-background. Strongly suggest changing to signal-to-background ratio to be consistent with the terminology used in the field.

We agree with the reviewer that this is confusing. We have defined SBR now as signal-to-background to improve clarity of conformity to current literature.

2. It isn't clear why cetuximab was selected for EGFR targeting vs. panitumumab since panitumumab is known to have an improved safety profile. Indeed adverse events upon infusion with cetuximab were seen. Please provide detail on this antibody choice for the study.

In the current study, we have chosen to use cetuximab-800CW instead of panitumumab-800CW due to our extensive experience with cetuximab-800CW. This project started in 2015 prior to the development of panatumimab-800CW. Developing panatumimab-800CW as a tracer would mean a completely new start of the project with a lot of effort and costs in GMP-production, building a new IMPD, dosing strategy and so on. We agree that panitumumab has an improved safety profile as shown by Eben Rosenthal's group¹, but we expect that a head-to-head comparison of the two imaging agents would most likely show similar tumor uptake and thus comparable results.

3. The pre-dosing strategy with unlabeled antibody and its utility should be covered in the introduction section.

We thank the reviewer for addressing this point. We have removed the rationale of the pre-dosing strategy from the Methods section to the Introduction section:

(Lines 86-88) "Multiple phase I feasibility studies have demonstrated the safety of EGFR-targeting tracers such as cetuximab-800CW and panitumumab-800CW, and their potential for real-time intraoperative margin assessment by showing fluorescent spots (possible lesions) in the deep margin²⁴⁻²⁶. In these studies, pre-dosing with unlabelled antibody prior to tracer administration showed improved contrast, most likely by preventing rapid plasma clearance of the tracer and occupying off-target receptors in normal tissue. To date, no well-powered phase II studies to evaluate the diagnostic accuracy of EGFR-targeted FMI for intraoperative margin assessment have been published so far.

4. Why was the SBR cut off set at >1.5 for close margins, but at >2 for tumor-positive margin detection overall? This is not explained.

Our data shows that an SBR of >2 indicates a high risk of a tumor-positive margin, and all tumor-positive margins were detected when using a SBR of >2 as cut-off. Yet, as stated in the results, using this cut-off would result in low sensitivity for the detection of close margins (43%). Lowering the cut off to >1.5 will increase the sensitivity for close margin detection to 76%.

With the current dosing strategy, we advocate to use a cut-off of >1.5 to identify at-risk margins that should be further evaluated with a microscopic technique (e.g. frozen sections or other optical techniques) for confirmation of our fluorescence imaging results, as also explained in a recently published perspective paper². This would lead to high sensitivity for both tumor-positive and close margins, while avoiding the removal of healthy tissue unnecessarily. We also think that this cut-off might be different for a particular tracer influenced by the time interval and the tissue that is studied.

5. The standard of care for an R0 resection needs to be defined in the introduction for contextual understanding of the study parameters. It isn't mentioned in the current manuscript until the end of the methods section.

In the introduction we have provided the definition of tumor-positive and close margins. A R0 resection is defined as >5mm on H&E histopathology. We have added this to the introduction section:

(Lines 59-60) "Surgical resection is often the primary treatment, with the aim of complete tumor removal. In OSCC surgery, a complete resection (or R0 resection), is defined as a histological margin of >5 mm. Yet, inadequate margins (<5 mm) occur in OSCC surgery at one of the highest rates in surgical oncology, with tumor-positive (0-1 mm) margins occurring in up to 40% of cases and close margins (1-5 mm) in up to 45%, primarily located in the deep margin."

6. *What system was used to image in vivo in patients? It appears based on the methods this system may be the SurgVision Explorer Air, but it isn't clear. Please provide additional detail.*

The reviewer is correct that the SurgVision Explorer Air was used for *in vivo* imaging. We have clarified this in the Methods section:

(Line 299) "The SurgVision Explorer Air® (SurgVision GmbH, Munich, Germany) was used for *in vivo* imaging and benchmarked using a fluorescence phantom prior to surgery."

7. *Would this methodology always be used for "back table" imaging to quantify tumor margins or would it also be used for imaging the tumor within the patient quantitatively in its future clinical implementation?*

At this stage of clinical translation, we believe that *ex vivo* "back table" imaging is superior to *in vivo* imaging for margin assessment due to several reasons. First of all, *in vivo* imaging is constrained by regulatory concerns of medical devices inherent to a sterile working environment. Moreover, and more importantly, in the *in vivo* imaging environment it is challenging to ensure constant image-acquisition settings such as camera distance and angulation of the illumination to the surgical working field. The *ex vivo* imaging environment enables more rigorous standardization of imaging procedures.² Consequently, standardization and calibration protocols can be developed to establish benchmarking of camera systems and quality controls. Therefore, we think that the future of margin assessment will be in *ex vivo* analysis.

8. *Why is the sensitivity and specificity of the Pear-Trilogy generally better than the SurgVision Explorer Air system?*

Based on a different reviewer's comment, we have reperformed the statistical analyses comparing the diagnostic performance of the SurgVision Explorer Air and Pearl Trilogy. We have now considered clustering of data, and still the sensitivity and specificity show no significant differences for the imaging systems. We have adapted the manuscript accordingly:

(Lines 193-198) "In planes imaged by both the SurgVision Explorer Air® and the Pearl-Trilogy® we observed a Pearson correlation for clustered data of 0.87. AUCs of positive margins were not different between the SurgVision Explorer Air® and the Pearl® (AUC of 0.92 (95% CI 0.79 to 1.0), 0.99 (95% CI 0.97 to 1.0), bootstrap p=0.11, respectively), and neither for close margins (AUC of 0.75 (95% CI 0.64 to 0.86), 0.77 (95% 0.67 to 0.87), bootstrap p=0.48, respectively)."

Furthermore, in another study we have compared both systems using a composite fluorescence phantom developed by colleagues of the Technical University Munich.³ In that study, we have compared multiple fluorescence camera systems based on a variety of performance characteristics. Both the phantom and formalin-fixed tissue were used for analysis. The data of that study will be reported separately and thus cannot be added to the manuscript or supplementary data. For the reviewer, we would like to refer to the figures below, in which no difference was observed in sensitivity

and depth sensitivity between the SurgVision Explorer Air and the Li-COR Pearl Trilogy. Please note, this is pre-liminary, unpublished data.

For these measurements, liquid phantoms were used with concentrations of IRDye800CW (Li-COR Biosciences, Lincoln, NE, USA) dissolved in water. All concentrations of were placed in microtubes (Sarstedt, colourless microtube 1.5 mL, code 72.692, screw cap). The tubes, split into high (10 μ M, 1 μ M, 100nM, 0) and low (10 nM, 1 nM, 100 pM, 0) concentrations were placed in holders each holding 4 samples. During imaging with PEARL Trilogy the 800 channel (excitation: 785 nm; emission: 820 nm) was used without any other adjustable settings. During imaging with the SurgVision Explorer Air II exposure times of 25 ms, 50 ms, and 100 ms were used.

Figure 1: Pearl sensitivity

Figure 2: SurgVision sensitivity

Figure 3: Sensitivity vs. depth for PEARL (left) and SurgVision (right)

9. The caption on Supplementary Fig 3 isn't clear. Specifically, it is stated that "The detection rate for tumor-positive margins is 100% for specimens with (Panel A) and without bone (Panel B)." While the AUC of the ROC in Panel A appears to be 1.0, this isn't the case for Panel B. Please clarify.

The reviewer is correct that the AUC of Panel B is not 1.0. Yet, an AUC of 1.0 is not a prerequisite of obtaining 100% sensitivity but would imply perfect diagnostic performance with 100% sensitivity and 100% specificity. As shown by the red line in Panel B, a cut-off value that results in a sensitivity of 100%, would provide a specificity of approximately 75%. This would imply the detection of all tumor-positive margins with some margins being classified as false-positive.

References:

1. Gao R.W., Teraphongphom N., de Boer E. et al. Safety of panitumumab-IRDye800CW and cetuximab-IRDye800CW for fluorescence-guided surgical navigation in head and neck cancers. *Theranostics*, 2018
2. Voskuil F.J., Vonk J., van der Vegt B. et al. Intraoperative imaging in pathology-assisted surgery. *Nature biomedical engineering*, 2022

3. Gorpas D., Koch M., Anastasopoulou M. et al. Benchmarking of fluorescence cameras through the use of a composite phantom. *Journal of Biomedical Optics*, 2017

Reviewer #3 (Remarks to the Author): with expertise in fluorescence-guided surgery

The manuscript presented by de Wit et al, summarizes the clinical findings from a prospective, Phase II fluorescence molecular imaging trial of oral squamous cell carcinoma patients. The diagnostic accuracy of cetuximab-800CW to detect oral cancers was investigated 2 days after a 75 mg infusion of unlabelled cetuximab and 15 mg of cetuximab-800CW. Tumors were imaged in situ with the SurgVision Explorer Air and spectra were acquired with multi-diameter single-fibre reflectance, single fibre fluorescence spectroscopy to quantify cetuximab-800CW fluorescence. Ex vivo specimen imaged with the SurgVision Explorer Air and Pearl-Trilogy. Diagnostic assessment of the deep margin was performed in the operating room using spot-to-background ratios (SBR). The study resulted in high sensitivity (100%) and specificity (86%) for tumor-positive margins with an SBR equal to or greater than 2, and lower sensitivity (70%) and specificity (76%) for close margins using an SBR equal to or greater than 1.5. This work further validates the use of sentinel margin assessment to identify positive, or close, tumor margins in the surgical suite in real time. Overall, the study is well developed, and the results presented are strongly in favor of adding fluorescence molecular imaging to surgical resection of oral squamous cell carcinomas. Furthermore, the results provide quantifiable measures to guide the margin assessment that may result in significant improvement in patient outcomes. The clinical impact of this study and resulting Phase III trials could be significant in the reduction of adjuvant therapies and treatments. However, there are some clarifications that need to be addressed in terms of the image exclusion criteria, the ex vivo diagnostic accuracy results from the two imaging systems, and multivariate analysis of data (multiple images from the same resection).

Major comments:

1. *The positive results of the study are well documented and explained. However, as the team is moving towards Phase III clinical testing, the more scientifically interesting question, and potential to improve upon the current results in continued testing, is to investigate why some specimen were miscategorized. This is only briefly mentioned in lines 159-172 of the Results, but no data is shown.*

A. *There were 11 false negatives, with n=4 in the 1-3 mm margin and n=7 in the 3-5 mm margin. Only the 1-3 mm margin were explained (low number of cells with necrosis or inflammation). What about the 7 miscategorized 3-5 mm margin? Was this size dependent? EGFR expression? Can the region be correlated back to the specimen and the actual TBR be calculated?*

We thank the reviewer for pointing this out and have clarified this in the results.

(Lines 171-178) "Using an SBR of 1.5 as a cut-off value (below an SBR of 1.5 the observers did not detect fluorescent spots), 11 false-negative close margins (n=4 in the 1-3 mm margin width group and n=7 in the 3-5 mm margin group) were found. Three out of four false-negatives of 1-3 mm showed small tumors with a low number of viable tumor cells combined with necrosis (n=1) or extensive inflammation encompassing the tumor (n=2). For the remaining missed 3-5 mm margins (n=7), immunohistochemistry could not explain the false negative results. Since most unexplained missed margins occur in the 3-5 mm margin group, we surmise these are due to limited depth information of the current SBR approach."

Below we have included representative examples of false-negative and false-positive results. We prefer not to add these figures to the manuscript to keep the paper more concise. We can add the figures to the supplementary materials upon request by the reviewer or editor.

The figure above shows typical examples of false negative results. A) A false negative fluorescence result, corresponding to a close margin of 1.9mm. Note that on the H&E slice, the tumor consists mostly of a large necrotic core with limited viable tumor cells, thus little EGFR receptors. This explains the absence of fluorescence in the centre of the tumor that results in insufficient fluorescent signal to create a spot in the resection margin. B) A false negative margin of 2.2mm, for which immunohistochemistry cannot provide an explanation for the absence of fluorescence signal in the margin.

The figure above shows typical examples of false negative results. A) A false negative fluorescence result, corresponding to a close margin of 1.9mm. Note that on the H&E slice, the tumor consists mostly of a large necrotic core with limited viable tumor cells, thus little EGFR receptors. This explains the absence of fluorescence in the centre of the tumor that results in insufficient fluorescent signal to create a spot in the resection margin. B) A false negative margin of 2.2mm, for which immunohistochemistry cannot provide an explanation for the absence of fluorescence signal in the margin.

b. In the false-positive margins, there was n=5 patients that had salivary gland in the margin. However, in Fig. 2 salivary gland is listed as an exclusion criterion for 1 specimen. It is unclear how salivary gland is being treated, especially since the authors state that “the surgeon can determine the presence of salivary glands in the resection margin using visual and tactile information.” Were these salivary glands detected by the surgeon? Why were some excluded and others not? What if the salivary glands were involved?

In the case of salivary tissue being clinically identifiable intraoperatively, we excluded the image since here the surgeon could distinguish between salivary and surrounding tissue. In false positives mentioned in the text containing salivary glands (n=5), the salivary glands were not identified during surgery but only on histopathology. All these corresponded to fluorescent signal in the margin and in 3 we also showed fluorescence signal on cross-sectional fluorescent imaging. This means that they could be of influence on imaging results, however salivary gland tissue was not clinically evident to the surgeon, and these images were therefore not excluded from pre-analysis. We believe that this type

of false-positive margins based on fluorescence of salivary glands will be identified during a frozen section analysis.

The figure above shows typical examples of salivary gland tissue in the resected specimen. In A) the salivary gland tissue corresponds with fluorescence signal on cross-sectional imaging. In B) the salivary gland does not correspond with fluorescence signal on cross-sectional imaging.

2. Lines 301-306, Figure 2: Image exclusion criteria are not well defined in the procedures. The list of image exclusion criteria in Figure 2 do not match that in the procedures, lines 301-306. Salivary gland is excluded, as well as "scattering of mucosa".

a. Can "scattering of mucosa" be more well defined in the procedure? Why was this an issue in 5 patients in the SurgVision system and only 4 patients in the Pearl system?

We agree that "scattering of the mucosa" might be the wrong terminology. These cases were actually due to reflection of light in wet mucosa, mainly in "wet mucosa". We have clarified this in the Method section.

(Lines 326-327) "Excluded images either had the surface of the specimen not perpendicular to the camera, a margin only consisting of bone, intraoperative extra resections performed and attached to the specimen prior to fluorescence imaging, or reflection of light in mucosal tissue interfered with evaluation of the resection margin."

Due to the slight difference in angle of the camera to the specimen due to placement of the specimen, this problem occurred 5 times when using the SurgVision and 4 times when using the Pearl.

b. Please explain the salivary gland exclusion and why others were included.

See also our answer to question 1B. In the case of salivary tissue being clinically identifiable intraoperatively, we excluded the image since here the surgeon can then distinguish between salivary and surrounding tissue. In false positives mentioned in the text containing salivary glands (n=5), the salivary glands were not identified during surgery but only on histopathology. This means that they could be of influence on imaging results, however salivary gland tissue was not clinically evident to the surgeon, and these images were therefore not excluded from pre-analysis.

3. The spot-to-background (SBR) results appear to be a combination of images from the Pearl and the SurgVision systems; however, it is unclear what combination of these two systems are used.

We refer to the Methods section where we describe how both systems are used for analysis of SBR results. Both imaging devices were used for margin assessment, and in the case of a fluorescent lesion on both devices, the highest SBR was noted. We have added a statement to clarify this procedure:

(Lines 334-338): When a fluorescent spot was identified with only one of the devices, the SBR of this fluorescent spot was used for analysis. In case a fluorescent spot was found with both imaging devices, the highest SBR (i.e., from either the Pearl-Trilogy or the SurgVision Explorer Air) was used for definitive analysis to make sure that no at-risk margins were missed."

a. The SurgVision was better at determining the close margins than the Pearl system for both the SBR greater or equal to 1 and 1.5, however the results reported in Table 1, the abstract, and throughout the manuscript report higher values than each system alone. Please clarify since the combined results appear to be quite a bit higher.

We used a combination of both techniques. In some cases, for instance where a re-resection was performed intraoperatively, we could not perform Pearl imaging since this system was localized in the Department of Pathology instead of the OR. The surgeons suture the re-resected margin anatomically on the resection specimen, which impedes the possibility to image the initial margin with the Pearl. Therefore, the combined sensitivity and specificity is higher than for each imaging device individually.

b. My instinct is that one system is better at determining the close margins in the 3-5 mm range than the other. Is there correlation between SBR and depth for each system?

While the Pearl Trilogy is specifically built for *ex vivo* imaging, we did not observe a statistically significant difference with the SurgVision Explorer Air. We have reperformed the statistical analyses comparing the diagnostic performance of the SurgVision Explorer Air and Pearl Trilogy. We have now considered clustering of data, and still the sensitivity and specificity show no significant differences for the imaging systems. We have adapted the manuscript accordingly:

(Lines 193-198) "In planes imaged by both the SurgVision Explorer Air® and the Pearl-Trilogy® we observed a Pearson correlation for clustered data of 0.87. AUCs of positive margins were not different between the SurgVision Explorer Air® and the Pearl® (AUC of 0.92 (95% CI 0.79 to 1.0), 0.99 (95% CI

0.97 to 1.0), bootstrap p=0.11, respectively), and neither for close margins (AUC of 0.75 (95% CI 0.64 to 0.86), 0.77 (95% 0.67 to 0.87), bootstrap p=0.48, respectively).”

Furthermore, in another study we have compared both systems using a composite fluorescence phantom developed by colleagues of the Technical University Munich.³ In that study, we have compared multiple fluorescence camera systems based on a variety of performance characteristics. Both the phantom and formalin-fixed tissue were used for analysis. The data of that study will be reported separately and thus cannot be added to the manuscript or supplementary data. For the reviewer, we would like to refer to the figures below, in which no difference was observed in sensitivity and depth sensitivity between the SurgVision Explorer Air and the Pearl Trilogy.

For these measurements, liquid phantoms were used with concentrations of IRDye800CW (Li-COR Biosciences, Lincoln, NE, USA) dissolved in water. All concentrations of were placed in microtubes (Sarstedt, colourless microtube 1.5 mL, code 72.692, screw cap). The tubes, split into high (10 μM, 1 μM, 100nM, 0) and low (10 nM, 1 nM, 100 pM, 0) concentrations were placed in holders each holding 4 samples. During imaging with PEARL Trilogy the 800 channel (excitation: 785 nm; emission: 820 nm) was used without any other adjustable settings. During imaging with the SurgVision Explorer Air II exposure times of 25 ms, 50 ms, and 100 ms were used.

Figure 1: Pearl sensitivity

Figure 2: SurgVision sensitivity

Figure 3: Sensitivity vs. depth for PEARL (left) and SurgVision (right)

c. Is it clinically viable in the future to image every excised specimen with two systems? The Pearl is advantageous for positive margins, while the SurgVision is advantageous for close margins. Please discuss and provide analysis.

We did not find a significant difference in the performance of the systems. We think that *ex vivo* analysis is sufficient for margin assessment and the Pearl system seems well equipped for this. Imaging systems for *in vivo* imaging such as the SurgVision Explorer Air generally are more expensive and more

complex to use. Therefore, we think that eventually the Pearl system will prove to be sufficient as well as better affordable for reliable margin analysis.

4. How were the statistics handled for multiple images from the same patient? Is this considered when determining the diagnostic accuracy? The descriptions of the statistical analysis in the procedures are vague.

We have explained this in the Methods section based on Saha et al.⁴:

Lines 385-388: “We estimated margin-level sensitivity, specificity, and positive and negative predicted values and corresponding 95% confidence intervals using an ANOVA-type Wilson score estimation that considers clustering within patients.”

5. All fluorescence images would benefit from being scaled to the same level, instead of the color map being scaled from “high” to “low”. Are they all scaled with the same “high” and “low” value? Or individually scaled.

It is important that all fluorescence images are scaled to the same level. As variations exist between patients, we have scaled the images per patient so that the patient serves as its own internal control. In the field of fluorescence image analysis, it is accepted to scale the fluorescence from zero to the maximum fluorescence intensity observed. In other words, the maximum fluorescence intensity observed in the tumor, was used for all other images analyzed for that patient as usual in this field. We have changed the method sections accordingly to provide more clarity.

(Lines 328-329) “All images were scaled to the maximum fluorescence intensity observed in the tumor of each patient, as variations in tumor biology exist with each patient. A region of interest was drawn around each identified fluorescent spot on the excised specimen. A background region of interest was drawn on the same fluorescence image that included adjacent tissue of the same origin (e.g., connective tissue, muscle) without the fluorescent spot.”

6. How were multiple regions in the margin addressed? In lines 326-327, it is stated that if multiple regions in the margin were detected then all were included. However in Figure 4B, there is a close margin indicated on the larger circled tumor but there appears to be a satellite tumor region that is not addressed in terms of whether it is also a close margin. The fluorescence from second smaller region seems negligible. Addition of the measured SBR for these regions on the images would greatly help the reader interpret the results.

We agree that the tissue slice with the apparent satellite lesion might be confusing to the reader. However, this does not represent a satellite lesion but rather the same tumor growing submucosally, therefore an individual SBR is not given in this example. To avoid confusing the reader, we changed the tissue slice to the tissue slice adjacent to the current one and pointed out the small tumor deposit. We altered the figure in the manuscript accordingly, and think that this solves the issue sufficiently.

(Lines 573 – 575) **Figure 5: Representative examples of a tumor-positive margin, close margin and tumor-negative margin.** Typical examples of A) a tumor-positive margin, B) a close margin, and C) a tumor-negative margin. *In vivo* fluorescence imaging shows sharply demarcated tumors compared to adjacent tissue (upper left images), and after excision no fluorescence can be detected in the wound bed (upper right images). On the tissue slices, the tumor is delineated with a solid black line. Panel A shows a fluorescent spot with an SBR of 4.0 on the excised specimen, corresponding to a tumor-positive margin (red arrows). Panel B shows a fluorescence spot with an SBR of 2.3, revealing a close margin of 2.2 mm (red arrows). The yellow arrow indicates a fluorescent lesion in the mucosa, which corresponds to the tumor spreading mucosally. In panel C, no fluorescent signal is seen in the margin, corresponding to a tumor-negative margin.

Other comments:

1. Lines 43-44 – Is the sentence starting “Fluorescence molecular imaging (FMI) has been explored...” only talking about oral squamous cell carcinomas? It could be argued that FMI is used for margin/residual tumor in ovarian and lung cancer. This sentence just needs clarification that it is not globally speaking to all FMI.

We thank the reviewer for the comment and have changed the abstract accordingly.

(Line 43) “Fluorescence molecular imaging (FMI) has been explored for intraoperative margin assessment, but data for OSCC are limited to phase I studies.”

2. *What are the main differences between Supplementary Table 1 and Supplementary Table 2?*

We now realize that we have summarized the data per imaging system in these supplementary tables. Since not all specimens were imaged by both imaging systems, we suggest leaving these tables out as these will be confusing. We have amended the supplementary data accordingly.

a. Supplementary Table 1 is never called out in the text of the manuscript and only some of the values are reported in the abstract and in the text.

See our explanation below question 2.

b. It seems Supplementary Table 1 is the values determined as described in lines 311-317 (the highest SBR of the two systems)?

See our explanation below question 2.

c. Supplementary Table 1 summarizes the results for close margin 1-3 and 3-5 but not 1-5, which is the value reported in the abstract and in lines 135-137. These data should be included in Table 1

See our explanation below question 2.

d. The differentiation of close margins at 1-3 mm depth and 3-5 mm depth should also be included in Table 2.

See our explanation below question 2.

e. Please clarify how many images were used from each system for the Table 1.

See our explanation below question 2.

f. Please include the image numbers for all tables, as they varied between systems.

See our explanation below question 2.

3. *The language is confusing around the discussion of fluorescence ratios. The only ratio that is defined is the spot-to-background ratio (SBR) the following are also discussed.*

We agree that the various terms and abbreviations are confusing. We have updated the terminology also in response to a different reviewer's comment:

(Lines 359-360) The trial's primary endpoints were the FMI detection rate of tumor-positive surgical margins, and the cut-off value for SBR that can be used for intraoperative detection of tumor-positive margins. The secondary endpoints were the detection rates of close surgical margins, the in vivo fluorescence contrast between tumor and adjacent tissue as determined by MDSFR/SFF defined as $TBR_{\text{spectroscopy}}$, and the tolerability and safety of cetuximab-800CW, for which adverse events were graded according to Common Terminology Criteria for Adverse Events (CTCAE) version 5.0.

a. Tumor to normal mucosa ratio (line 199)

Tumor to normal mucosa ratio is only measured by MDSFR/SFF spectroscopy and is therefore changed to $TBR_{\text{spectroscopy}}$, please also see the answer above.

b. Median ratio (line 115)

This median relates to the median $TBR_{\text{spectroscopy}}$, please also see the answer to comment 3a.

c. TBR (Figure 1B, second y-axis, line 489)

TBR was changed to SBR (signal-to-background ratio) in response to a comment from a different reviewer. We believe it is now clear that SBR relates to fluorescence molecular imaging and $TBR_{\text{spectroscopy}}$ relates to spectroscopy or intrinsic fluorescence.

4. Figure 5 would benefit from a description on why the top figure of the figure is different than the bottom of the figure. Is it just location and number of surgical margins? Does this influence the number of images from the patient?

We thank the reviewer for this comment. We have clarified this in the description of the figure:

(Lines 543-545): “A) In vivo fluorescence imaging of the tumor. B) Back table imaging of the excised specimen. Fluorescence imaging is performed from all surgical planes of the specimen. In the case of a complex specimen, multiple surgical planes can be identified and imaged, and in the case of a simple specimen only one surgical plane per specimen is imaged. Fluorescent spots are observed in image 5 (top row) and image 1 (bottom row). C) Bread loaf slicing of the specimen and fluorescence imaging of all bread loaf slices. D) Correlation of the fluorescent spots relate to tumor-positive margins on histopathology.”

5. Was the phantom imaged on both the SurgVision and the Pearl systems? This is not stated but it appears that results from only one system are provided.

The phantom was only imaged for benchmarking the SurgVision camera system, since for this system the working distance (i.e. distance from the camera to the tissue of interest) had to be manually adjusted for each specimen due to size differences and could therefore benefit from phantom imaging. For the Pearl, the working distance is fixed. Next to this we checked the sensitivity of both systems which was not different for both devices..

6. Lines 208-210 – The statement “Previously, we showed that predosing with unlabelled cetuximab can block off-target receptors, thus increasing tumor uptake of the fluorescent tracer and reducing the background signal.” is a very strong statement to make and is not adequately demonstrated in Reference 27. The data in Reference 27 shows that the fluorescence normal surrounding tissue stays relatively constant in most doses (excluding the 50 mg cetuximab-800CW) in Figure 4B and 4C. The additional images provided in Figure S6 do not adequately demonstrate significant off-target signal reduction due to receptor saturation. You do not histologically demonstrate that the signal outside of the tumor ROIs is actually from EGFR in normal tissue and you are providing representative data, not entire cohorts analyzed.

The reviewer is correct that this statement might be too strong, and we have adjusted this accordingly. We have shown that the signal-to-background ratio improves when performing pre-dosing². Colleagues have shown that co-administration of an unlabelled antibody improves penetration of the antibody-drug conjugate³.

We have adapted the sentence as follows:

(Lines 225-228) “Previously, we showed that pre-dosing with unlabelled cetuximab results in increased contrast between tumor and normal tissue. This may be the result of preventing rapid plasma clearance of the tracer and occupying off-target receptors in normal tissue.”

References:

1. Gorpas D., Koch M., Anastasopoulou M. et al. Benchmarking of fluorescence cameras through the use of a composite phantom. *Journal of Biomedical Optics*, 2017
2. Voskuil, F. J. *et al.* Fluorescence-guided imaging for resection margin evaluation in head and neck cancer patients using cetuximab-800CW: A quantitative dose-escalation study. *Theranostics* 10, 3994–4005 (2020).
3. Lu G., van den Berg N., Brock M.A. et al. Co-administered antibody improves penetration of antibody–dye conjugate into human cancers with implications for antibody–drug conjugates. *Nature communications*, 2020.
4. Saha, K. K., Miller, D. & Wang, S. A Comparison of Some Approximate Confidence Intervals for a Single Proportion for Clustered Binary Outcome Data. *Int J Biostat* 12, 20150024 s(2016).

Reviewer #4 (Remarks to the Author): with expertise in biostatistics, clinical trial study design
This review pertains only to the statistical aspect of this manuscript.

The authors identified several cut-off values in spot-to-background ratio (SBR) for predicting several binary endpoints. Sensitivity, specificity, positive and negative predictive values, as well as the receiving operating characteristic (ROC) curves as well as area under ROC curves (AUC) are provided for each binary endpoint. The optimal cut-off values are determined by Youden's Index (i.e., maximum sensitivity + specificity). Understandably, most of the numbers and curves (Figure 3, B-D) are not ideal due to the limited sample size.

The reviewer would suggest performing a recursive partitioning regression model, in addition to the existing statistical methods. This is only a suggestion, not a requirement.

We appreciate the suggestion of the reviewer to adopt multivariable analyses to explore whether the combination of SBR (the primary variable we report on now) with other fluorescence signal read-outs (e.g. maximum fluorescence intensity of the spot) and possibly also other information sources for instance related to the surgical specimen to predict margin status would be worthwhile. For this recursive partitioning (or methods such as ridge regression or XGBoost) would indeed be an interesting option. Nevertheless, we refrained from doing so given the small dataset at hand with only 14 tumor-positive margins, precluding meaningful multivariable analyses in our opinion. Furthermore, the SBR in our dataset already yields an AUC of 0.95 for positive margins, leaving little room for improvement. For close margins the situation is somewhat different with an SBR AUC of 0.72. Here we have 37 “events”: still not enough to meaningfully develop and explore the performance of a multivariable model.

With the argumentation above we hope to have reasonably explained why we chose not to pursue any multivariable analyses. As we will proceed to use our technique in new patients, we foresee to expand our data which may allow for such analyses in the future.

REVIEWERS' COMMENTS

Reviewer #1 (Remarks to the Author):

Thank you for responding to my queries and suggestions. I feel that most of the response are satisfactory and I appreciate the clarification in the manuscript. I feel that the following should be included in the manuscript or supplementary material (which were mentioned in the rebuttal letter) as they are important for completeness:

1. The actual surgical protocol (important positives and negatives in the practice)...unfortunately there is no such thing as standard surgical practice as there is significant variation as to how much of a gross margin one plans for up front, and how this is determined (estimation versus ruler), use of neck dissection /SNB and frozen sections. These should be categorically stated so that the results can be contextualized accordingly.
2. The false positive and false negative images shows should also be in the supplementary data
3. Survival stats should also be in the manuscript/supplementary, as these are important to contextualise your study population.

Reviewer #5 (Remarks to the Author):

The authors have responded sufficiently to critiques from reviewer 2 and reviewer 3. In all cases the changes in the manuscript are warranted and helpful. In just one case where supplemental data and figure is shown on false negative examples, this information should be added to the supplementary materials. This also supports the critique number 4 from reviewer 1 which was concerned about false negatives and false positives. Further insight was afforded by these figures as additions to supplementary information, and along with clinical outcomes.

Point-by-point Response "*Fluorescence molecular imaging for intraoperative margin assessment in oral cancer 2 patients: a prospective, phase II imaging trial*".

Jaron G. de Wit, Jasper Vonk, Floris J. Voskuil, Sebastiaan A.H.J. de Visscher, KeesPieter Schepman, Wouter T.R. Hooghiemstra, Matthijs D. Linssen, Sjoerd G. Elias, Gyorgy B. Halmos, Boudewijn E.C. Plaat, Jan J. Doff, Eben L. Rosenthal, Dominic Robinson, Bert van der Vegt, Wouter B. Nagengast, Gooitzen M. van Dam, Max J.H. Witjes

We would like to thank the reviewers again for their constructive feedback. We have provided a point-by-point response to the remaining issues and have altered the manuscript where applicable.

The reviewer's comments are stated in *italic*. Our responses are written in **blue font color**. Cited paragraphs from the revised manuscript are displayed in **brown font color**. New or rewritten sections in these sentences are underlined. In our updated manuscript, we highlighted amended sections with yellow background and additional sections with green background.

REVIEWERS' COMMENTS

Reviewer #1 (Remarks to the Author):

Thank you for responding to my queries and suggestions. I feel that most of the response are satisfactory and I appreciate the clarification in the manuscript. I feel that the following should be included in the manuscript or supplementary material (which were mentioned in the rebuttal letter) as they are important for completeness:

1. The actual surgical protocol (important positives and negatives in the practice)...unfortunately there is no such thing as standard surgical practice as there is significant variation as to how much of a gross margin one plans for up front, and how this is determined (estimation versus ruler), use of neck dissection /SNB and frozen sections. These should be categorically stated so that the results can be contextualized accordingly.

We have elaborated on the standard of care surgery as performed in our hospital in the methods section as described below.

"Two days after tracer administration, surgery was performed according standard of care in our hospital. Routinely, CT and/or MRI were preoperatively available, on which the primary resection was planned. The tumor was removed with an estimated clinical margin of 1cm. Fresh frozen sectioning of the margins was not performed routinely. Concomitant neck dissections or sentinel node biopsies were performed based on cTN classification, according to Dutch guidelines. In the case of a clinical suspicion of an irradical resection intraoperatively, an extra resection was performed and attached to the primary specimen in the same setting."

2. The false positive and false negative images shows should also be in the supplementary data

We have provided figures showing representative examples of false positive and false negative results the supplemental material (Supplementary figures 2 & 3).

3. Survival stats should also be in the manuscript/supplementary, as these are important to contextualise your study population.

We have added this information in the supplemental data.

Reviewer #5 (Remarks to the Author):

The authors have responded sufficiently to critiques from reviewer 2 and reviewer 3. In all cases the changes in the manuscript are warranted and helpful. In just one case where supplemental data and figure is shown on false negative examples, this information should be added to the supplementary materials. This also supports the critique number 4 from reviewer 1 which was concerned about false negatives and false positives. Further insight was afforded by these figures as additions to supplementary information, and along with clinical outcomes.

We have provided figures showing representative examples of false positive and false negative results the supplemental material (Supplementary figures 2 & 3).